# SAdam: A Variant of Adam for Strongly Convex Functions

**Guanghui Wang[1], Shiyin Lu[1], Quan Cheng[1], Wei-Wei Tu[2] and Lijun Zhang[1,\*]**
[1]National Key Laboratory for Novel Software Technology, Nanjing University, China
[2]4Paradigm Inc., Beijing, China
{wanggh,lusy,chengq,zhanglj}@lamda.nju.edu.cn,tuwwcn@gmail.com

## Abstract

The Adam algorithm has become extremely popular for large-scale machine learning. Under convexity condition, it has been proved to enjoy a data-dependent $O(\sqrt{T})$ regret bound where $T$ is the time horizon. However, whether *strong convexity* can be utilized to further improve the performance remains an open problem. In this paper, we give an affirmative answer by developing a variant of Adam (referred to as *SAdam*) which achieves a data-dependent $O(\log T)$ regret bound for strongly convex functions. The essential idea is to maintain a faster decaying yet under controlled step size for exploiting strong convexity. In addition, under a special configuration of hyperparameters, our SAdam reduces to SC-RMSprop, a recently proposed variant of RMSprop for strongly convex functions, for which we provide the *first* data-dependent logarithmic regret bound. Empirical results on optimizing strongly convex functions and training deep networks demonstrate the effectiveness of our method.

## 1 Introduction

Online Convex Optimization (OCO) is a well-established learning framework which has both theoretical and practical appeals (Shalev-Shwartz et al., 2012). It is performed in a sequence of consecutive rounds: In each round $t$, firstly a learner chooses a decision $\mathbf{x}_t$ from a convex set $\mathcal{D} \subseteq \mathbb{R}^d$, at the same time, an adversary reveals a loss function $f_t(\cdot) : \mathcal{D} \mapsto \mathbb{R}$, and consequently the learner suffers a loss $f_t(\mathbf{x}_t)$. The goal is to minimize regret, defined as the difference between the cumulative loss of the learner and that of the best decision in hindsight (Hazan et al., 2016):

$$R(T) := \sum_{t=1}^{T} f_t(\mathbf{x}_t) - \min_{\mathbf{x} \in \mathcal{D}} \sum_{t=1}^{T} f_t(\mathbf{x}).$$

The most classic algorithm for OCO is Online Gradient Descent (OGD) (Zinkevich, 2003), which attains an $O(\sqrt{T})$ regret. OGD iteratively performs descent step towards gradient direction with a *predetermined* step size, which is oblivious to the characteristics of the data being observed. As a result, its regret bound is *data-independent*, and can not benefit from the structure of data. To address this limitation, various of *adaptive* gradient methods, such as Adagrad (Duchi et al., 2011), RMSprop (Tieleman & Hinton, 2012) and Adadelta (Zeiler, 2012) have been proposed to exploit the geometry of historical data. Among them, Adam (Kingma & Ba, 2015), which dynamically adjusts the step size and the update direction by exponential average of the past gradients, has been extensively popular and successfully applied to many applications (Xu et al., 2015; Gregor et al., 2015; Kiros et al., 2015; Denkowski & Neubig, 2017; Bahar et al., 2017). Despite the outstanding performance, Reddi et al. (2018) pointed out that Adam suffers the *non-convergence* issue, and developed two modified versions, namely AMSgrad and AdamNC. These variants are equipped with *data-dependent* regret bounds, which are $O(\sqrt{T})$ in the worst case and become tighter when gradients are sparse [1].

---

[\*]Lijun Zhang is the corresponding author.

[1]We note that, as very recently pointed out by Tran et al. (2019), there still exists a minor theoretical flaw in the analysis of Reddi et al. (2018), and such issue in fact appears in many of recent variants of AMSgrad/AdamNC. In this paper, we provide a simple way to fix this problem. The details can be found in Appendix G.

While the theoretical behavior of Adam in convex cases becomes clear, it remains an open problem whether *strong convexity* can be exploited to achieve better performance. Such property arises, for instance, in support vector machines as well as other regularized learning problems, and it is well-known that the vanilla OGD with an appropriately chosen step size enjoys a much better $O(\log T)$ regret bound for strongly convex functions (Hazan et al., 2007). In this paper, we propose a variant of Adam adapted to strongly convex functions, referred to as *SAdam*. Our algorithm follows the general framework of Adam, yet keeping a faster decaying step size controlled by time-variant heperparameters to exploit strong convexity. Theoretical analysis demonstrates that SAdam achieves a data-dependent $O(\log T)$ regret bound for strongly convex functions, which means that it converges faster than AMSgrad and AdamNC in such cases, and also enjoys a huge gain in the face of sparse gradients.

Furthermore, under a special configuration of heperparameters, the proposed algorithm reduces to the SC-RMSprop (Mukkamala & Hein, 2017), which is a variant of RMSprop algorithm for strongly convex functions. We provide an alternative proof for SC-RMSprop, and establish the *first* data-dependent logarithmic regret bound. Finally, we evaluate the proposed algorithm on strongly convex problems as well as deep networks, and the empirical results demonstrate the effectiveness of our method.

**Notation.** Throughout the paper, we use lower case bold face letters to denote vectors, lower case letters to denote scalars, and upper case letters to denote matrices. We use $\| \cdot \|$ to denote the $\ell_2$-norm and $\| \cdot \|_\infty$ the infinite norm. For a positive definite matrix $H \in \mathbb{R}^{d \times d}$, the weighted $\ell_2$-norm is defined by $\|\mathbf{x}\|_H^2 = \mathbf{x}^\top H \mathbf{x}$. The $H$-weighted projection $\Pi_\mathcal{D}^H(\mathbf{x})$ of $\mathbf{x}$ onto $\mathcal{D}$ is defined by $\Pi_\mathcal{D}^H(\mathbf{x}) = \arg\min_{\mathbf{y} \in \mathcal{D}} \|\mathbf{y} - \mathbf{x}\|_H^2$. We use $\mathbf{g}_t$ to denote the gradient of $f_t(\cdot)$ at $\mathbf{x}_t$. For vector sequence $\{\mathbf{v}_t\}_{t=1}^T$, we denote the $i$-th element of $\mathbf{v}_t$ by $v_{t,i}$. For diagonal matrix sequence $\{M_t\}_{t=1}^T$, we use $m_{t,i}$ to denote the $i$-th element in the diagonal of $M_t$. We use $\mathbf{g}_{1:t,i} = [g_{1,i}, \cdots, g_{t,i}]$ to denote the vector obtained by concatenating the $i$-th element of the gradient sequence $\{\mathbf{g}_t\}_{t=1}^T$.

## 2 RELATED WORK

In this section, we briefly review related work in online convex and strongly convex optimization.

In the literature, most studies are devoted to the minimization of regret for convex functions. Under the assumptions that the infinite norm of gradients and the diameter of the decision set are bounded, OGD with step size on the order of $O(1/\sqrt{t})$ (referred to as convex OGD) achieves a data-independent $O(d\sqrt{T})$ regret (Zinkevich, 2003), where $d$ is the dimension. To conduct more informative updates, Duchi et al. (2011) introduce Adagrad algorithm, which adjusts the step size of OGD in a per-dimension basis according to the geometry of the past gradients. In particular, the diagonal version of the algorithm updates decisions as

$$\mathbf{x}_{t+1} = \mathbf{x}_t - \frac{\alpha}{\sqrt{t}} V_t^{-1/2} \mathbf{g}_t \tag{1}$$

where $\alpha > 0$ is a constant factor, $V_t$ is a $d \times d$ diagonal matrix, and $\forall i \in [d], v_{t,i} = (\sum_{j=1}^t g_{j,i}^2)/t$ is the arithmetic average of the square of the $i$-th elements of the past gradients. Intuitively, while the step size of Adagrad, i.e., $(\alpha/\sqrt{t})V_t^{-1/2}$, decreases generally on the order of $O(1/\sqrt{t})$ as that in convex OGD, the additional matrix $V_t^{-1/2}$ will automatically increase step sizes for sparse dimensions in order to seize the infrequent yet valuable information therein. For convex functions, Adagrad enjoys an $O(\sum_{i=1}^d \|\mathbf{g}_{1:T,i}\|)$ regret, which is $O(d\sqrt{T})$ in the worst case and becomes tighter when gradients are sparse.

Although Adagrad works well in sparse cases, its performance has been found to deteriorate when gradients are dense due to the rapid decay of the step size since it uses all the past gradients in the update (Zeiler, 2012). To tackle this issue, Tieleman & Hinton (2012) propose RMSprop, which alters the arithmetic average procedure with Exponential Moving Average (EMA), i.e.,

$$V_t = \beta V_{t-1} + (1 - \beta)\text{diag}(\mathbf{g}_t \mathbf{g}_t^\top)$$

where $\beta \in [0, 1]$ is a hyperparameter, and $\text{diag}(\cdot)$ denotes extracting the diagonal matrix. In this way, the weights assigned to past gradients decay exponentially so that the reliance of the update is

essentially limited to recent few gradients. Since the invention of RMSprop, many EMA variants of Adagrad have been developed (Zeiler, 2012; Kingma & Ba, 2015; Dozat, 2016). One of the most popular algorithms is Adam (Kingma & Ba, 2015), where the first-order momentum acceleration, shown in (2), is incorporated into RMSprop to boost the performance:

$$\hat{\mathbf{g}}_t = \beta_1 \hat{\mathbf{g}}_{t-1} + (1 - \beta_1)\mathbf{g}_t \tag{2}$$

$$V_t = \beta_2 V_{t-1} + (1 - \beta_2)\mathrm{diag}(\mathbf{g}_t \mathbf{g}_t^\top) \tag{3}$$

$$\mathbf{x}_{t+1} = \mathbf{x}_t - \frac{\alpha}{\sqrt{t}} V_t^{-1/2} \hat{\mathbf{g}}_t \tag{4}$$

While it has been successfully applied to various practical applications, a recent study by Reddi et al. (2018) shows that Adam could fail to converge to the optimal decision even in some simple one-dimensional convex scenarios due to the potential rapid fluctuation of the step size. To resolve this issue, they design two modified versions of Adam. The first one is AMSgrad,

$$\hat{V}_t = \max\left\{\hat{V}_{t-1}, V_t\right\}$$

$$\mathbf{x}_{t+1} = \mathbf{x}_t - \frac{\alpha}{\sqrt{t}} \hat{V}_t^{-1/2} \hat{\mathbf{g}}_t$$

where an additional element-wise maximization procedure is employed before the update of $\mathbf{x}_t$ to ensure a stable step size. The other is AdamNC, where the framework of Adam remains unchanged, yet a time-variant $\beta_2$ (i.e., $\beta_{2t}$) is adopted to keep the step size under control. Theoretically, the two algorithms achieve data-dependent $O(\sqrt{T}\sum_{i=1}^d v_{T,i} + \sum_{i=1}^d \|\mathbf{g}_{1:T,i}\|\log T)$ and $O(\sqrt{T}\sum_{i=1}^d v_{T,i} + \sum_{i=1}^d \|\mathbf{g}_{1:T,i}\|)$ regrets respectively. In the worst case, they suffer $O(d\sqrt{T}\log T)$ and $O(d\sqrt{T})$ regrets respectively, and enjoy a huge gain when data is sparse.

Note that the aforementioned algorithms are mainly analysed in general convex settings and suffer at least $O(d\sqrt{T})$ regret in the worst case. For online strongly convex optimization, the classical OGD with step size proportional to $O(1/t)$ (referred to as strongly convex OGD) achieves a data-independent $O(d\log T)$ regret (Hazan et al., 2007). Inspired by this, Mukkamala & Hein (2017) modify the update rule of Adagrad in (1) as follows

$$\mathbf{x}_{t+1} = \mathbf{x}_t - \frac{\alpha}{t} V_t^{-1} \mathbf{g}_t$$

so that the step size decays approximately on the order of $O(1/t)$, which is similar to that in strongly convex OGD. The new algorithm, named SC-Adagrad, is proved to enjoy a data-dependent regret bound of $O(\sum_{i=1}^d \log(\|\mathbf{g}_{1:T,i}\|^2))$, which is $O(d\log T)$ in the worst case. They further extend this idea to RMSprop, and propose an algorithm named SC-RMSprop. However, as pointed out in Section 3, their regret bound for SC-RMSprop is in fact *data-independent*, and in this paper we provide the first data-dependent regret bound for this algorithm.

Very recently, several modifications of Adam adapted to non-convex settings have been developed (Chen et al., 2019; Basu et al., 2018; Zhang et al., 2018; Shazeer & Stern, 2018). However, to our knowledge, none of these algorithms are particularly designed for strongly convex functions, nor enjoy a logarithmic regret bound.

## 3 SADAM

In this section, we first describe the proposed algorithm, then state its theoretical guarantees, and finally compare it with the SC-RMSprop algorithm.

### 3.1 THE ALGORITHM

Before proceeding to our algorithm, following previous studies, we introduce some standard definitions (Boyd & Vandenberghe, 2004) and assumptions (Reddi et al., 2018).

**Definition 1.** *A function $f(\cdot) : \mathcal{D} \mapsto \mathbb{R}$ is $\lambda$-strongly convex if $\forall \mathbf{x}_1, \mathbf{x}_2 \in \mathcal{D}$,*

$$f(\mathbf{x}_1) \geq f(\mathbf{x}_2) + \nabla f(\mathbf{x}_2)^\top (\mathbf{x}_1 - \mathbf{x}_2) + \frac{\lambda}{2}\|\mathbf{x}_1 - \mathbf{x}_2\|^2. \tag{5}$$

---

**Algorithm 1** SAdam

---

1: **Input:** $\{\beta_{1t}\}_{t=1}^T, \{\beta_{2t}\}_{t=1}^T, \delta$
2: **Initialize:** $\hat{\mathbf{g}}_0 = \mathbf{0}$, $\hat{V}_0 = \mathbf{0}_{d\times d}$, $\mathbf{x}_1 = \mathbf{0}$.
3: **for** $t = 1, \ldots, T$ **do**
4:     $\mathbf{g}_t = \nabla f_t(\mathbf{x}_t)$
5:     $\hat{\mathbf{g}}_t = \beta_{1t}\hat{\mathbf{g}}_{t-1} + (1 - \beta_{1t})\mathbf{g}_t$
6:     $V_t = \beta_{2t}V_{t-1} + (1 - \beta_{2t})\text{diag}(\mathbf{g}_t\mathbf{g}_t^\top)$
7:     $\hat{V}_t = V_t + \frac{\delta}{t}I_d$
8:     $\mathbf{x}_{t+1} = \Pi_{\mathcal{D}}^{\hat{V}_t}\left(\mathbf{x}_t - \frac{\alpha}{t}\hat{V}_t^{-1}\hat{\mathbf{g}}_t\right)$
9: **end for**

---

**Assumption 1.** *The infinite norm of the gradients of all loss functions are bounded by $G_\infty$, i.e., their exists a constant $G_\infty > 0$ such that $\max_{\mathbf{x}\in\mathcal{D}} \|\nabla f_t(\mathbf{x})\|_\infty \leq G_\infty$ holds for all $t \in [T]$.*

**Assumption 2.** *The decision set $\mathcal{D}$ is bounded. Specifically, their exists a constant $D_\infty > 0$ such that $\max_{\mathbf{x}_1,\mathbf{x}_2\in\mathcal{D}} \|\mathbf{x}_1 - \mathbf{x}_2\|_\infty \leq D_\infty$.*

We are now ready to present our algorithm, which follows the general framework of Adam and is summarized in Algorithm 1. In each round $t$, we firstly observe the gradient at $\mathbf{x}_t$ (Step 4), then compute the first-order momentum $\hat{\mathbf{g}}_t$ (Step 5). Here $\beta_{1t}$ a time-variant hyperparameter. Next, we calculate the second-order momentum $V_t$ by EMA of the square of past gradients (Step 6). This procedure is controlled by $\beta_{2t}$, whose value will be discussed later. After that, we add a vanishing factor $\frac{\delta}{t}$ to the diagonal of $V_t$ and get $\hat{V}_t$ (Step 7), which is a standard technique for avoiding too large steps caused by small gradients in the beginning iterations. Finally, we update the decision by $\hat{\mathbf{g}}_t$ and $\hat{V}_t$, which is then projected onto the decision set (Step 8).

While SAdam is inspired by Adam, there exist two key differences: One is the update rule of $\mathbf{x}_t$ in Step 8, and the other is the configuration of $\beta_{2t}$ in Step 6. Intuitively, both modifications stem from strongly convex OGD, and jointly result in a faster decaying yet under controlled step size which helps utilize the strong convexity while preserving the practical benefits of Adam. Specifically, in the first modification, we remove the square root operation in (4) of Adam, and update $\mathbf{x}_t$ at Step 8 as follows

$$\mathbf{x}_{t+1} = \mathbf{x}_t - \frac{\alpha}{t}\hat{V}_t^{-1}\hat{\mathbf{g}}_t. \tag{6}$$

In this way, the step size used to update the $i$-th element of $\mathbf{x}_t$ is $\frac{\alpha}{t}\hat{v}_{t,i}^{-1}$, which decays in general on the order of $O(1/t)$, and can still be automatically tuned in a per-feature basis via the EMA of the historical gradients.

The second modification is made to $\beta_{2t}$, which determines the value of $V_t$ and thus also controls the decaying rate of the step size. To help understand the motivation behind our algorithm, we first revisit Adam, where $\beta_{2t}$ is simply set to be constant, which, however, could cause rapid fluctuation of the step size, and further leads to the non-convergence issue. To ensure convergence, Reddi et al. (2018) propose that $\beta_{2t}$ should satisfy the following two conditions:

**Condition 1.** $\forall t \in [T]$ *and* $i \in [d]$,

$$\frac{\sqrt{t}v_{t,i}^{1/2}}{\alpha} - \frac{\sqrt{t-1}v_{t-1,i}^{1/2}}{\alpha} \geq 0.$$

**Condition 2.** *For some $\zeta > 0$ and $\forall t \in [T]$, $i \in [d]$,*

$$\sqrt{t\sum_{j=1}^t \Pi_{k=1}^{t-j}\beta_{2(t-k+1)}(1-\beta_{2j})g_{j,i}^2} \geq \frac{1}{\zeta}\sqrt{\sum_{j=1}^t g_{j,i}^2}.$$

The first condition implies that the difference between the inverses of step sizes in two consecutive rounds is positive. It is inherently motivated by convex OGD (i.e., OGD with step size $\frac{\alpha}{\sqrt{t}}$, where $\alpha > 0$ is a constant factor), in which

$$\frac{\sqrt{t}}{\alpha} - \frac{\sqrt{t-1}}{\alpha} \geq 0, \forall t \in [T]$$

is a key condition used in the analysis. We first modify Condition 1 by mimicking the behavior of strongly convex OGD as we are devoted to minimizing regret for strongly convex functions. In strongly convex OGD (Hazan et al., 2007), the step size at each round $t$ is set as $\frac{\alpha}{t}$ with $\alpha \geq \frac{1}{\lambda}$ for $\lambda$-strongly convex functions. Under this configuration, we have

$$\frac{t}{\alpha} - \frac{t-1}{\alpha} \leq \lambda, \forall t \in [T]. \tag{7}$$

Motivated by this, we propose the following condition for our SAdam, which is an analog to (7).

**Condition 3.** *Their exists a constant $C > 0$ such that for any $\alpha \geq \frac{C}{\lambda}$, we have $\forall t \in [T]$ and $i \in [d]$,*

$$\frac{tv_{t,i}}{\alpha} - \frac{(t-1)v_{t-1,i}}{\alpha} \leq \lambda(1 - \beta_1). \tag{8}$$

Note that the extra $(1 - \beta_1)$ in the righthand side of (8) is necessary because SAdam involves the first-order momentum in its update.

Finally, since the step size of SAdam scales with $1/t$ rather than $1/\sqrt{t}$ in Adam, we modify Condition 2 accordingly as follows:

**Condition 4.** *For some $\zeta > 0$, $\forall t \in [T]$ and $i \in [d]$,*

$$t \sum_{j=1}^{t} \prod_{k=1}^{t-j} \beta_{2(t-k+1)}(1 - \beta_{2j})g_{j,i}^2 \geq \frac{1}{\zeta} \sum_{j=1}^{t} g_{j,i}^2. \tag{9}$$

## 3.2 THEORETICAL GUARANTEES

In the following, we give a general regret bound when the two conditions are satisfied.

**Theorem 1.** *Suppose Assumptions 1 and 2 hold, and all loss functions $f_1(\cdot), \ldots, f_T(\cdot)$ are $\lambda$-strongly convex. Let $\delta > 0$, $\beta_{1t} = \beta_1 \nu^{t-1}$, where $\beta_1 \in [0,1)$, $\nu \in [0,1)$, and $\{\beta_{2t}\}_{t=1}^{T} \in [0,1]^T$ be a parameter sequence such that Conditions 3 and 4 are satisfied. Let $\alpha \geq \frac{C}{\lambda}$. The regret of SAdam satisfies*

$$R(T) \leq \frac{dD_{\infty}^2 \delta}{2\alpha(1 - \beta_1)} + \frac{d\beta_1 D_{\infty}^2 (G_{\infty}^2 + \delta)}{2\alpha(1 - \beta_1)(\nu - 1)^2} + \frac{\alpha\zeta}{(1 - \beta_1)^3} \sum_{i=1}^{d} \log\left(\frac{1}{\zeta\delta} \sum_{j=1}^{T} g_{j,i}^2 + 1\right). \tag{10}$$

**Remark 1.** The above theorem implies that our algorithm enjoys an $O(\sum_{i=1}^{d} \log(\|\mathbf{g}_{1:T,i}\|^2))$ regret bound, which is $O(d \log T)$ in the worst case, and automatically becomes tighter whenever the gradients are small or sparse such that $\|\mathbf{g}_{1:T,i}\|^2 \ll G_{\infty}^2 T$ for some $i \in [d]$. The superiority of data-dependent bounds have been witnessed by a long list of literature, such as Duchi et al. (2011); Mukkamala & Hein (2017); Reddi et al. (2018). In the following, we give some concrete examples:

- Consider a one-dimensional sparse setting where non-zero gradient appears with probability $c/T$ and $c > 1$ is a constant. Then $\mathbb{E}\left[\log\left(\sum_{t=1}^{T} g_{t,1}^2\right)\right] = O(\log(c))$, which is a constant factor.

- Consider a high-dimensional sparse setting where in each dimension of gradient non-zero element appears with probability $p = T^{(m-d)/d}$ with $m \in [1, d)$ being a constant. Then, $\mathbb{E}\left[\sum_{i=1}^{d} \log\left(\sum_{j=1}^{T} g_{j,i}^2\right)\right] = O(m \log T)$, which is much tighter than $O(d \log T)$.

**Remark 2.** In practice, first-order momentum is a powerful technique that can significantly boost the performance (Reddi et al., 2018), and our paper is the first to show that algorithms equipped with such technique can achieve logarithmic regret bound for strongly convex functions. However, since the regret bound of SAdam is data-dependent, it is difficult to rigorously analyse the influence of the first-order momentum parameter $\beta_1$ as it affects all the gradients appearing in the last term of the regret of Theorem 1. We will further investigate this in the feature work. We note that the regret bounds of adaptive algorithms with first-order momentum (e.g., Reddi et al., 2018; Chen et al., 2019) all share a similar structure as our regret bound with respect to $\beta_1$.

Next, we provide an instantiation of $\{\beta_{2t}\}_{t=1}^{T}$ such that Conditions 3 and 4 hold, and derive the following Corollary.

**Corollary 2.** *Suppose Assumptions 1 and 2 hold, and all loss functions $f_1(\cdot), \ldots, f_T(\cdot)$ are $\lambda$-strongly convex. Let $\delta > 0$, $\beta_{1t} = \beta_1 \nu^{t-1}$ where $\nu, \beta_1 \in [0, 1)$, and $1 - \frac{1}{t} \leq \beta_{2t} \leq 1 - \frac{\gamma}{t}$, where $\gamma \in (0, 1]$. Then we have:*

*1. For any $\alpha \geq \frac{(2-\gamma)G_\infty^2}{\lambda(1-\beta_1)}$, $\forall t \in [T]$ and $i \in [d]$,*

$$\frac{t v_{t,i}}{\alpha} - \frac{(t-1)v_{t-1,i}}{\alpha} \leq \lambda(1-\beta_1).$$

*2. For all $t \in [T]$ and $j \in [d]$,*

$$t \sum_{j=1}^{t} \prod_{k=1}^{t-j} \beta_{2(t-k+1)}(1-\beta_{2j})g_{j,i}^2 \geq \gamma \sum_{j=1}^{t} g_{j,i}^2.$$

*Moreover, let $\alpha \geq \frac{(2-\gamma)G_\infty^2}{\lambda(1-\beta_1)}$, and the regret of SAdam satisfies:*

$$R(T) \leq \frac{dD_\infty^2 \delta}{2\alpha(1-\beta_1)} + \frac{d\beta_1 D_\infty^2 (G_\infty^2 + \delta)}{2\alpha(1-\beta_1)(\nu-1)^2} + \frac{\alpha}{\gamma(1-\beta_1)^3} \sum_{i=1}^{d} \log\left(\frac{\gamma}{\delta}\sum_{j=1}^{T} g_{j,i}^2 + 1\right).$$

Furthermore, as a special case, by setting $\beta_{1t} = 0$ and $1 - \frac{1}{t} \leq \beta_{2t} \leq 1 - \frac{\gamma}{t}$, our algorithm reduces to SC-RMSprop (Mukkamala & Hein, 2017), which is a variant of RMSprop for strongly convex functions. Although Mukkamala & Hein (2017) have provided theoretical guarantees for this algorithm, we note that their regret bound is in fact *data-independent*. Specifically, the regret bound provided by Mukkamala & Hein (2017) takes the following form:

$$R(T) \leq \frac{\delta d D_\infty^2}{2\alpha} + \frac{\alpha}{2\gamma}\sum_{i=1}^{d}\log\left(\frac{T v_{T,i}}{\delta} + 1\right) + \frac{\alpha}{\gamma}\sum_{i=1}^{d}\frac{(1-\gamma)(1+\log T)}{\inf_{j\in[1,T]} j v_{j,i} + \delta}. \tag{11}$$

Focusing on the denominator of the last term in (11), we have

$$\inf_{j\in[1,T]} j v_{j,i} + \delta \leq 1 v_{1,i} + \delta \leq G_\infty + \delta$$

thus

$$\frac{\alpha}{\gamma}\sum_{i=1}^{d}\frac{(1-\gamma)(1+\log T)}{\inf_{j\in[1,T]} j v_{j,i} + \delta} \geq \frac{d\alpha}{\gamma}\frac{(1-\gamma)(\log T + 1)}{G_\infty + \delta}$$

which implies that their regret bound is of order $O(d \log T)$, and thus data-independent. In contrast, based on Corollary 2, we present a new regret bound for SC-RMSprop in the following, which is $O(\sum_{i=1}^{d}\log(\|\mathbf{g}_{1:T,i}\|^2))$, and thus data-dependent.

**Corollary 3.** *Suppose Assumptions 1 and 2 hold, and all loss functions $f_1(\cdot), \ldots, f_T(\cdot)$ are $\lambda$-strongly convex. Let $\delta > 0$, $\beta_{1t} = 0$, and $1 - \frac{1}{t} \leq \beta_{2t} \leq 1 - \frac{\gamma}{t}$, where $\gamma \in (0, 1]$. Let $\alpha \geq \frac{(2-\gamma)G_\infty^2}{\lambda}$. Then SAdam reduces to SC-RMSprop, and its regret satisfies*

$$R(T) \leq \frac{dD_\infty^2 \delta}{2\alpha} + \frac{\alpha}{\gamma}\sum_{i=1}^{d}\log\left(\frac{\gamma}{\delta}\sum_{j=1}^{T} g_{j,i}^2 + 1\right). \tag{12}$$

Finally, we note that Mukkamala & Hein (2017) also consider a more general version of SC-RMSprop which uses a time-variant non-increasing $\delta$ for each dimension $i$. In Appendix D we introduce the $\delta$-variant technique to our SAdam, and provide the corresponding theoretical guarantee.

## 4 EXPERIMENTS

In this section, we present empirical results on optimizing strongly convex functions and training deep networks. More results can be found Appendix.

**Algorithms.** In both experiments, we compare the following algorithms:

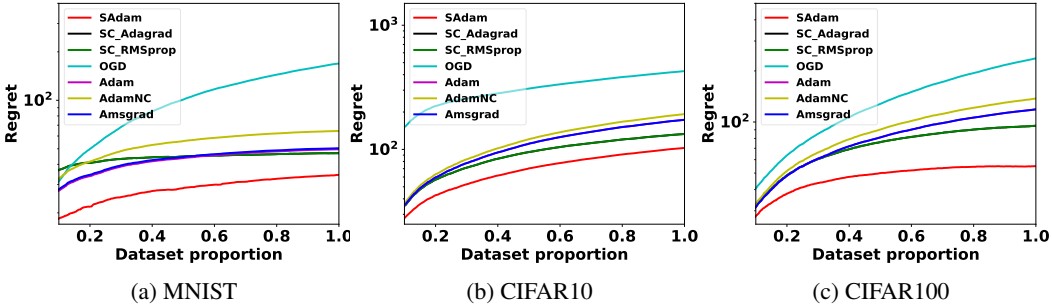

(a) MNIST            (b) CIFAR10            (c) CIFAR100

Fig. 1: Regret v.s. data proportion for $\ell_2$-regularized softmax regression

- SC-Adagrad (Mukkamala & Hein, 2017), with step size $\alpha_t = \alpha/t$.
- SC-RMSprop (Mukkamala & Hein, 2017), with step size $\alpha_t = \alpha/t$ and $\beta_t = 1 - \frac{0.9}{t}$.
- Adam (Kingma & Ba, 2015) and AMSgrad (Reddi et al., 2018), both with $\beta_1 = 0.9$, $\beta_2 = 0.999$, $\alpha_t = \alpha/\sqrt{t}$ for convex problems and time-invariant $\alpha_t = \alpha$ for non-convex problems.
- AdamNC (Reddi et al., 2018), with $\beta_1 = 0.9$, $\beta_{2t} = 1 - 1/t$, and $\alpha_t = \alpha/\sqrt{t}$ for convex problems and a time-invariant $\alpha_t = \alpha$ for non-convex problems.
- Online Gradient Descent (OGD), with step size $\alpha_t = \alpha/t$ for strongly convex problems and a time-invariant $\alpha_t = \alpha$ for non-convex problems.
- Our proposed SAdam, with $\beta_1 = 0.9$, $\beta_{2t} = 1 - \frac{0.9}{t}$.

For Adam, AdamNC and AMSgrad, we choose $\delta = 10^{-8}$ according to the recommendations in their papers. For SC-Adagrad and SC-RMSprop, following Mukkamala & Hein (2017), we choose a time-variant $\delta_{t,i} = \xi_2 e^{-\xi_1 t v_{t,i}}$ for each dimension $i$, with $\xi_1 = 0.1$, $\xi_2 = 1$ for convex problems and $\xi_1 = 0.1$, $\xi_2 = 0.1$ for non-convex problems. For our SAdam, since the removing of the square root procedure and very small gradients may cause too large step sizes in the beginning iterations, we use a rather large $\delta = 10^{-2}$ to avoid this problem. To conduct a fair comparison, for each algorithm, we choose $\alpha$ from the set $\{0.1, 0.01, 0.001, 0.0001\}$ and report the best results.

**Datasets.** In both experiments, we examine the performances of the aforementioned algorithms on three widely used datasets: MNIST (60000 training samples, 10000 test samples), CIFAR10 (50000 training samples, 10000 test samples), and CIFAR100 (50000 training samples, 10000 test samples). We refer to LeCun (1998) and Krizhevsky (2009) for more details of the three datasets.

### 4.1 OPTIMIZING STRONGLY CONVEX FUNCTIONS

In the first experiment, we consider the problem of mini-batch $\ell_2$-regularized softmax regression, which belongs to the online strongly convex optimization framework. Let $K$ be the number of classes and $m$ be the batch size. In each round $t$, firstly a mini-batch of training samples $\{(\mathbf{x}_m, y_m)\}_{i=1}^m$ arrives, where $y_i \in [K], \forall i \in [m]$. Then, the algorithm predicts parameter vectors $\{\mathbf{w}_i, b_i\}_{i=1}^K$, and suffers a loss which takes the following form:

$$J(\mathbf{w}, b) = -\frac{1}{m} \sum_{i=1}^m \log \left( \frac{e^{\mathbf{w}_{y_i}^\top \mathbf{x}_i + b_{y_i}}}{\sum_{j=1}^K e^{\mathbf{w}_j^\top \mathbf{x}_i + b_j}} \right) + \lambda_1 \sum_{k=1}^K \|\mathbf{w}_k\|^2 + \lambda_2 \sum_{k=1}^K b_k^2.$$

The value of $\lambda_1$ and $\lambda_2$ are set to be $10^{-2}$ for all experiments. The regret (in log scale) v.s. dataset proportion is shown in Fig. 1. It can be seen that our SAdam outperforms other methods across all the considered datasets. Besides, we observe that data-dependent strongly convex methods such as SC-Adagrad, SC-RMSprop and SAdam preform better than algorithms for general convex functions such as Adam, AMSgrad and AdamNC. Finally, OGD has the overall highest regret on all three datasets.

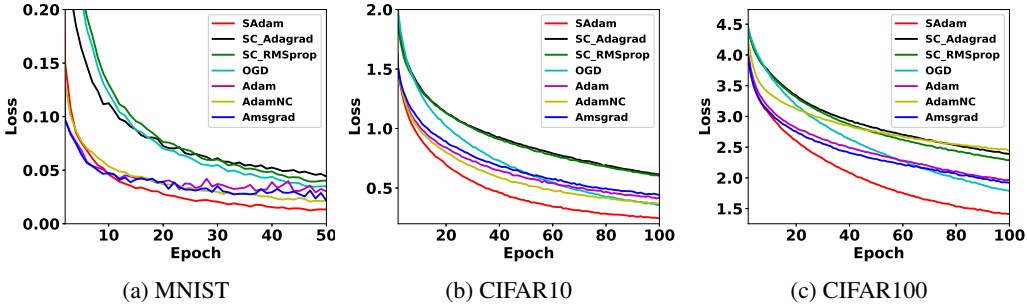

Fig. 2: Training loss v.s. number of epochs for 4-layer CNN

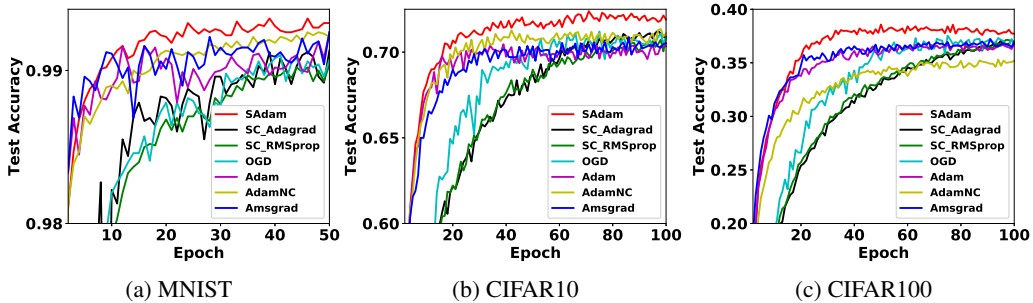

Fig. 3: Testing accuracy v.s. number of epochs for 4-layer CNN

## 4.2 TRAINING DEEP NETWORKS

Following Mukkamala & Hein (2017), we also conduct experiments on a 4-layer CNN, which consists of two convolutional layers (each with 32 filters of size $3 \times 3$), one max-pooling layer (with a $2 \times 2$ window and 0.25 dropout), and one fully connected layer (with 128 hidden units and 0.5 dropout). We employ ReLU function as the activation function for convolutional layers and softmax function as the activation function for the fully connected layer. The loss function is the cross-entropy. The training loss v.s. epoch is shown in Fig. 2, and the testing accuracy v.s. epoch is presented in Fig. 4. As can be seen, our SAdam achieves the lowest training loss on the three data sets. Moreover, this performance gain also translates into good performance on testing accuracy. The experimental results show that although our proposed SAdam is designed for strongly convex functions, it could lead to superior practical performance even in some highly non-convex cases such as deep learning tasks.

## 5 CONCLUSION AND FUTURE WORK

In this paper, we provide a variant of Adam adapted to strongly convex functions. The proposed algorithm, namely SAdam, follows the general framework of Adam, while keeping a step size decaying in general on the order of $O(1/t)$ and controlled by data-dependent heperparameters to exploit strong convexity. Theoretical analysis shows that SAdam achieves a data-dependent $O(d \log T)$ regret bound for strongly convex functions, which means that it converges much faster than Adam, AdamNC, and AMSgrad in such cases, and can enjoy a huge gain in the face of sparse gradients. In addition, we also provide the first data-dependent logarithmic regret bound for SC-RMSprop. Finally, we test the proposed algorithm on optimizing strongly convex functions as well as training deep networks, and the empirical results demonstrate the effectiveness of our method.

Since SAdam enjoys a data-dependent $O(d \log T)$ regret for online strongly convex optimization, it can be easily translated into a data-dependent $O(d \log T/T)$ convergence rate for *stochastic* strongly convex optimization (SSCO) by using the online-to-batch conversion (Kakade & Tewari, 2009). However, this rate is not optimal for SSCO, and it is sill an open problem how to achieve a data-dependent $O(d/T)$ convergence rate for SSCO. Recent development on adaptive gradient method (Chen et al., 2018) has proved that Adagrad combined with the multi-stage scheme (Hazan & Kale, 2014) can achieve this rate, but it is highly non-trivial to extend this technique to SAdam, and we leave it as a future work.

## 6 ACKNOWLEDGEMENT

This work was partially supported by NSFC (61976112), NSFC-NRF Joint Research Project (61861146001), and the Collaborative Innovation Center of Novel Software Technology and Industrialization.

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

# A    PROOF OF THEOREM 1

From Definition 1, we can upper bound regret as

$$R(T) = \sum_{t=1}^{T} f_t(\mathbf{x}_t) - \sum_{t=1}^{T} f_t(\mathbf{x}_*) \overset{(5)}{\leq} \sum_{t=1}^{T} \mathbf{g}_t^{\top} (\mathbf{x}_t - \mathbf{x}_*) - \frac{\lambda}{2} \|\mathbf{x}_t - \mathbf{x}_*\|^2 \tag{13}$$

where $\mathbf{x}_* := \min_{\mathbf{x} \in \mathcal{D}} \sum_{t=1}^{T} f_t(\mathbf{x})$ is the best decision in hindsight. On the other hand, by the update rule of $\mathbf{x}_{t+1}$ in Algorithm 1, we have

$$\begin{aligned}
\|\mathbf{x}_{t+1} - \mathbf{x}_*\|_{\hat{V}_t}^2 &= \left\| \Pi_{\mathcal{D}}^{\hat{V}_t} \left( \mathbf{x}_t - \alpha_t \hat{V}_t^{-1} \hat{\mathbf{g}}_t \right) - \mathbf{x}_* \right\|_{\hat{V}_t}^2 \\
&\leq \|\mathbf{x}_t - \alpha_t \hat{V}_t^{-1} \hat{\mathbf{g}}_t - \mathbf{x}_*\|_{\hat{V}_t}^2 \\
&= -2\alpha_t \hat{\mathbf{g}}_t^{\top} (\mathbf{x}_t - \mathbf{x}_*) + \|\mathbf{x}_t - \mathbf{x}_*\|_{\hat{V}_t}^2 + \alpha_t^2 \|\hat{\mathbf{g}}_t\|_{\hat{V}_t^{-1}}^2 \\
&= -2\alpha_t \left( \beta_{1t} \hat{\mathbf{g}}_{t-1} + (1 - \beta_{1t}) \mathbf{g}_t \right)^{\top} (\mathbf{x}_t - \mathbf{x}_*) \\
&\quad + \|\mathbf{x}_t - \mathbf{x}_*\|_{\hat{V}_t}^2 + \alpha_t^2 \|\hat{\mathbf{g}}_t\|_{\hat{V}_t^{-1}}^2
\end{aligned} \tag{14}$$

where $\alpha_t := \alpha/t$, and the inequality is due to the following lemma, which implies that the weighted projection procedure is non-expansive.

**Lemma 1.** *(McMahan & Streeter, 2010) Let $A \in \mathbb{R}^{d \times d}$ be a positive definite matrix and $\mathcal{D} \subseteq \mathbb{R}^d$ be a convex set. Then we have, $\forall \mathbf{x}, \mathbf{y} \in \mathbb{R}^d$,*

$$\|\Pi_{\mathcal{D}}^{A}(\mathbf{x}) - \Pi_{\mathcal{D}}^{A}(\mathbf{y})\|_A \leq \|\mathbf{x} - \mathbf{y}\|_A. \tag{15}$$

Rearranging (14), we have

$$\begin{aligned}
\mathbf{g}_t^{\top} (\mathbf{x}_t - \mathbf{x}_*) \leq & \frac{\|\mathbf{x}_t - \mathbf{x}_*\|_{\hat{V}_t}^2 - \|\mathbf{x}_{t+1} - \mathbf{x}_*\|_{\hat{V}_t}^2}{2\alpha_t (1 - \beta_{1t})} + \frac{\beta_{1t}}{1 - \beta_{1t}} \hat{\mathbf{g}}_{t-1}^{\top} (\mathbf{x}_* - \mathbf{x}_t) \\
& + \frac{\alpha_t}{2(1 - \beta_{1t})} \|\hat{\mathbf{g}}_t\|_{\hat{V}_t^{-1}}^2 \\
\leq & \frac{\|\mathbf{x}_t - \mathbf{x}_*\|_{\hat{V}_t}^2 - \|\mathbf{x}_{t+1} - \mathbf{x}_*\|_{\hat{V}_t}^2}{2\alpha_t (1 - \beta_{1t})} + \frac{\beta_{1t}}{1 - \beta_{1t}} \hat{\mathbf{g}}_{t-1}^{\top} (\mathbf{x}_* - \mathbf{x}_t) \\
& + \frac{\alpha_t}{2(1 - \beta_1)} \|\hat{\mathbf{g}}_t\|_{\hat{V}_t^{-1}}^2
\end{aligned} \tag{16}$$

where the last inequality is due to $\beta_{1t} \leq \beta_1$. We proceed to bound the second term of the inequality above. When $t \geq 2$, by Young's inequality and the definition of $\beta_{1t}$, we get

$$\begin{aligned}
\frac{\beta_{1t}}{1 - \beta_{1t}} \hat{\mathbf{g}}_{t-1}^{\top} (\mathbf{x}_* - \mathbf{x}_t) \leq & \frac{\alpha_{t-1} \beta_{1t}}{2(1 - \beta_{1t})} \|\hat{\mathbf{g}}_{t-1}\|_{\hat{V}_{t-1}^{-1}}^2 + \frac{\beta_{1t}}{2\alpha_{t-1}(1 - \beta_{1t})} \|\mathbf{x}_t - \mathbf{x}_*\|_{\hat{V}_{t-1}}^2 \\
\leq & \frac{\alpha_{t-1}}{2(1 - \beta_1)} \|\hat{\mathbf{g}}_{t-1}\|_{\hat{V}_{t-1}^{-1}}^2 + \frac{\beta_{1t}}{2\alpha_{t-1}(1 - \beta_{1t})} \|\mathbf{x}_t - \mathbf{x}_*\|_{\hat{V}_{t-1}}^2.
\end{aligned} \tag{17}$$

When $t = 1$, this term becomes 0 since $\hat{\mathbf{g}}_0 = \mathbf{0}$ in Algorithm 1. Plugging (16) and (17) into (13), we get the following inequality, of which we divide the righthand side into three parts and upper bound each of them one by one.

$$\begin{aligned}
R(T) \leq & \underbrace{\sum_{t=1}^{T} \left( \frac{\|\mathbf{x}_t - \mathbf{x}_*\|_{\hat{V}_t}^2 - \|\mathbf{x}_{t+1} - \mathbf{x}_*\|_{\hat{V}_t}^2}{2\alpha_t (1 - \beta_{1t})} - \frac{\lambda}{2} \|\mathbf{x}_t - \mathbf{x}_*\|^2 \right)}_{P_1} \\
& + \underbrace{\frac{1}{2(1 - \beta_1)} \sum_{t=2}^{T} \alpha_{t-1} \|\hat{\mathbf{g}}_{t-1}\|_{\hat{V}_{t-1}^{-1}}^2 + \frac{\sum_{t=1}^{T} \alpha_t \|\hat{\mathbf{g}}_t\|_{\hat{V}_t^{-1}}^2}{2(1 - \beta_1)}}_{P_2} + \underbrace{\sum_{t=2}^{T} \frac{\beta_{1t}}{2\alpha_{t-1}(1 - \beta_{1t})} \|\mathbf{x}_t - \mathbf{x}_*\|_{\hat{V}_{t-1}}^2}_{P_3}.
\end{aligned}$$

To bound $P_1$, we have

$$
\begin{aligned}
P_1 =& \frac{\|\mathbf{x}_1 - \mathbf{x}_*\|_{\hat{V}_1}^2}{2\alpha_1(1 - \beta_1)} \underbrace{- \frac{\|\mathbf{x}_{T+1} - \mathbf{x}_*\|_{V_T}^2}{2\alpha_T(1 - \beta_{1T})}}_{\leq 0} - \sum_{t=1}^{T} \left( \frac{\lambda}{2} \|\mathbf{x}_t - \mathbf{x}_*\|^2 \right) \\
&+ \sum_{t=2}^{T} \left( \frac{\|\mathbf{x}_t - \mathbf{x}_*\|_{\hat{V}_t}^2}{2\alpha_t(1 - \beta_{1t})} - \frac{\|\mathbf{x}_t - \mathbf{x}_*\|_{\hat{V}_{t-1}}^2}{2\alpha_{t-1}(1 - \beta_{1(t-1)})} \right) \\
\leq& \frac{\|\mathbf{x}_1 - \mathbf{x}_*\|_{\hat{V}_1}^2}{2\alpha_1(1 - \beta_1)} - \sum_{t=1}^{T} \left( \frac{\lambda}{2} \|\mathbf{x}_t - \mathbf{x}_*\|^2 \right) + \sum_{t=2}^{T} \frac{1}{1 - \beta_{1t}} \left( \frac{\|\mathbf{x}_t - \mathbf{x}_*\|_{\hat{V}_t}^2}{2\alpha_t} - \frac{\|\mathbf{x}_t - \mathbf{x}_*\|_{\hat{V}_{t-1}}^2}{2\alpha_{t-1}} \right) \\
=& \sum_{t=2}^{T} \frac{1}{2\alpha(1 - \beta_{1t})} \left( t\|\mathbf{x}_t - \mathbf{x}_*\|_{\hat{V}_t}^2 - (t-1)\|\mathbf{x}_t - \mathbf{x}_*\|_{\hat{V}_{t-1}}^2 - \lambda\alpha(1 - \beta_{1t})\|\mathbf{x}_t - \mathbf{x}_*\|^2 \right) \\
&+ \left( \frac{\|\mathbf{x}_1 - \mathbf{x}_*\|_{\hat{V}_1}^2}{2\alpha_1(1 - \beta_1)} - \frac{\lambda}{2} \|\mathbf{x}_1 - \mathbf{x}_*\|^2 \right)
\end{aligned}
\tag{18}
$$

where the inequality is derived from $\beta_{1(t-1)} \geq \beta_{1t}$. For the first term in the last equality of (18), we have

$$
\begin{aligned}
& t\|\mathbf{x}_t - \mathbf{x}_*\|_{\hat{V}_t}^2 - (t-1)\|\mathbf{x}_t - \mathbf{x}_*\|_{\hat{V}_{t-1}}^2 - \lambda\alpha(1 - \beta_{1t})\|\mathbf{x}_t - \mathbf{x}_*\|^2 \\
=& \sum_{i=1}^{d} (x_{t,i} - x_{*,i})^2 (t\hat{v}_{t,i} - (t-1)\hat{v}_{t-1,i} - \lambda\alpha(1 - \beta_{1t})) \\
=& \sum_{i=1}^{d} (x_{t,i} - x_{*,i})^2 (tv_{t,i} - (t-1)v_{t-1,i} - \lambda\alpha(1 - \beta_{1t})) \\
\overset{(8)}{\leq}& \sum_{i=1}^{d} (x_{t,i} - x_{*,i})^2 (\alpha\lambda(1 - \beta_1) - \lambda\alpha(1 - \beta_{1t})) \leq 0
\end{aligned}
\tag{19}
$$

where the second equality is because $\hat{v}_{t,i} = v_{t,i} + \frac{\delta}{t}$, and the second inequality is due to $1 - \beta_1 \leq 1 - \beta_{1t}$.
For the second term of (18), we have

$$
\begin{aligned}
\left( \frac{\|\mathbf{x}_1 - \mathbf{x}_*\|_{\hat{V}_1}^2}{2\alpha_1(1 - \beta_1)} - \frac{\lambda}{2} \|\mathbf{x}_1 - \mathbf{x}_*\|^2 \right) &= \sum_{i=1}^{d} (x_{1,i} - x_{*,i})^2 \left( \frac{\hat{v}_{1,i} - \lambda\alpha(1 - \beta_1)}{2\alpha(1 - \beta_1)} \right) \\
&= \sum_{i=1}^{d} (x_{1,i} - x_{*,i})^2 \left( \frac{v_{1,i} - \lambda\alpha(1 - \beta_1) + \delta}{2\alpha(1 - \beta_1)} \right) \\
&\leq \sum_{i=1}^{d} (x_{1,i} - x_{*,i})^2 \left( \frac{\delta}{2\alpha(1 - \beta_1)} \right) \\
&\leq \frac{dD_\infty^2 \delta}{2\alpha(1 - \beta_1)}
\end{aligned}
\tag{20}
$$

where first inequality is due to Condition 3, and the second inequality follows from Assumption 2.
Combining (18), (19) and (20), we get

$$
P_1 \leq \frac{dD_\infty^2 \delta}{2\alpha(1 - \beta_1)}.
\tag{21}
$$

To bound $P_2$, we introduce the following lemma.

**Lemma 2.** *The following inequality holds*

$$
\sum_{t=1}^{T} \alpha_t \|\hat{\mathbf{g}}_t\|_{\hat{V}_t^{-1}}^2 \leq \frac{\alpha\zeta}{(1 - \beta_1)^2} \sum_{i=1}^{d} \log \left( \frac{1}{\zeta\delta} \sum_{j=1}^{T} g_{j,i}^2 + 1 \right).
\tag{22}
$$

By Lemma 2, we have

$$
\begin{aligned}
P_2 &= \frac{1}{2(1-\beta_1)} \sum_{t=2}^{T} \alpha_{t-1} \|\hat{\mathbf{g}}_{t-1}\|_{\hat{V}_{t-1}^{-1}}^2 + \frac{\sum_{t=1}^{T} \alpha_t \|\hat{\mathbf{g}}_t\|_{\hat{V}_t^{-1}}^2}{2(1-\beta_1)} \\
&\leq \frac{1}{(1-\beta_1)} \sum_{t=1}^{T} \alpha_t \|\hat{\mathbf{g}}_t\|_{\hat{V}_t^{-1}}^2 \\
&\overset{(22)}{\leq} \frac{\alpha\zeta}{(1-\beta_1)^3} \sum_{i=1}^{d} \log\left( \frac{1}{\zeta\delta} \sum_{j=1}^{T} g_{j,i}^2 + 1 \right).
\end{aligned}
\tag{23}
$$

Finally, we turn to upper bound $P_3$:

$$
\begin{aligned}
P_3 &= \sum_{t=2}^{T} \frac{\beta_{1t}}{2\alpha_{t-1}(1-\beta_{1t})} \|\mathbf{x}_t - \mathbf{x}_*\|_{\hat{V}_{t-1}}^2 \\
&= \sum_{i=1}^{d} \sum_{t=2}^{T} \frac{\beta_{1t}}{2\alpha(1-\beta_{1t})} (x_{t,i} - x_{*,i})^2 (t-1)\hat{v}_{t-1,i} \\
&\leq \frac{D_\infty^2 (G_\infty^2 + \delta)}{2\alpha} \sum_{i=1}^{d} \sum_{t=1}^{T} \frac{\beta_{1t}}{1-\beta_{1t}} t \\
&\leq \frac{\beta_1 D_\infty^2 (G_\infty^2 + \delta)}{2\alpha} \sum_{i=1}^{d} \sum_{t=1}^{T} \frac{\nu^{t-1}}{1-\beta_1} t \\
&= \frac{\beta_1 D_\infty^2 (G_\infty^2 + \delta)}{2\alpha(1-\beta_1)} \sum_{i=1}^{d} \underbrace{\sum_{t=0}^{T-1} \nu^t (t+1)}_{P_3'}.
\end{aligned}
$$

To further bound $P_3'$, following Bock et al. (2019), we have

$$
\begin{aligned}
P_3' &= \sum_{t=0}^{T-1} \nu^t t + \nu^t \\
&= \left( \frac{(T-1)\nu^{T+1} - T\nu^T + \nu}{(\nu-1)^2} + \frac{1-\nu^T}{1-\nu} \right) \\
&= \frac{1 - T(\nu^T - \nu^{T+1}) - \nu T}{(1-\nu)^2} \\
&\leq \frac{1}{(1-\nu)^2}
\end{aligned}
\tag{24}
$$

where the inequality follows from $\nu^T \geq \nu^{T+1}$. Thus,

$$
P_3 \leq \frac{d\beta_1 D_\infty^2 (G_\infty^2 + \delta)}{2\alpha(1-\beta_1)(\nu-1)^2}.
\tag{25}
$$

We complete the proof by combining (21), (23) and (25).

## B    PROOF OF COROLLARY 2

For the first condition, we have

$$
\begin{aligned}
tv_{t,i} - (t-1)v_{t-1,i} &= t\beta_{2t} v_{t-1,i} + t(1-\beta_{2t})g_{t,i}^2 - (t-1)v_{t-1,i} \\
&\leq t\left(1 - \frac{\gamma}{t}\right) v_{t-1,i} + t\frac{1}{t}g_{t,i}^2 - (t-1)v_{t-1,i} \\
&\leq (t - \gamma - (t-1))v_{t-1,i} + G_\infty^2 \\
&\leq (2-\gamma)G_\infty^2
\end{aligned}
\tag{26}
$$

where the first inequality is derived from the definition of $\beta_{2t}$, the second and the third inequalities are due to Assumption 1. Based on (26), for any $\alpha \geq \frac{(2-\gamma)G_\infty^2}{\lambda(1-\beta_1)}$, we have $\frac{tv_{t,i}}{\alpha} - \frac{(t-1)v_{t-1,i}}{\alpha} \leq \lambda(1-\beta_1)$ holds for all $t \in [T]$ and $i \in [d]$.

For the second condition, we have

$$
\begin{aligned}
t\sum_{j=1}^{t}\prod_{k=1}^{t-j}\beta_{2(t-k+1)}\left(1-\beta_{2j}\right)g_{j,i}^2 &\geq t\sum_{j=1}^{t}\prod_{k=1}^{t-j}\left(1-\frac{1}{t-k+1}\right)\frac{\gamma}{j}g_{j,i}^2 \\
&= t\sum_{j=1}^{t}\prod_{k=1}^{t-j}\frac{t-k}{t-k+1}\frac{\gamma}{j}g_{j,i}^2 \\
&= t\sum_{j=1}^{t}\frac{j}{t}\frac{\gamma}{j}g_{j,i}^2 \\
&= \gamma\sum_{j=1}^{t}g_{j,i}^2
\end{aligned}
\tag{27}
$$

where the inequality follows from $\beta_t \geq 1 - \frac{1}{t}$ and $1 - \beta_t \geq \frac{\gamma}{t}$.

## C   PROOF OF LEMMA 2

We begin with the following lemma that is central to our analysis.

**Lemma 3.** *For all $i \in [d]$ and $t \in [T]$, we have*

$$
\sum_{j=1}^{T}\frac{g_{j,i}^2}{\sum_{k=1}^{j}g_{k,i}^2+\zeta\delta} \leq \log\left(\frac{\sum_{j=1}^{T}g_{j,i}^2}{\zeta\delta}+1\right).
\tag{28}
$$

*Proof.* For any $a \geq b > 0$, the inequality $1 + x \leq e^x$ implies that

$$
\frac{1}{a}(a-b) \leq \log\frac{a}{b}.
\tag{29}
$$

Let $m_0 = \zeta\delta$, and $m_j = \sum_{k=1}^{j}g_{k,i}^2 + \zeta\delta > 0$. By (29), we have

$$
\frac{g_{j,i}^2}{\sum_{k=1}^{j}g_{k,i}^2+\zeta\delta} = \frac{m_j - m_{j-1}}{m_j} \leq \log\frac{m_j}{m_{j-1}}.
$$

Summing over 1 to $T$, we have

$$
\sum_{j=1}^{T}\frac{g_{j,i}^2}{\sum_{k=1}^{j}g_{k,i}^2+\zeta\delta} \leq \log\frac{m_t}{m_0} = \log\left(\frac{\sum_{j=1}^{T}g_{j,i}^2}{\zeta\delta}+1\right).
$$

$\square$

Now we turn to the proof of Lemma 2. First, expending the last term in the summation by the update rule of Algorithm 1, we get

$$
\alpha_T\|\hat{\mathbf{g}}_T\|_{V_T^{-1}}^2 = \alpha_T\sum_{i=1}^{d}\frac{\hat{g}_{T,i}^2}{v_{T,i}+\frac{\delta}{T}} = \alpha\sum_{i=1}^{d}\frac{\left(\sum_{j=1}^{T}(1-\beta_{1j})\prod_{k=1}^{T-j}\beta_{1(T-k+1)}g_{j,i}\right)^2}{T\sum_{j=1}^{T}(1-\beta_{2j})\Pi_{k=1}^{T-j}\beta_{2(T-k+1)}g_{j,i}^2+\delta}.
\tag{30}
$$

The above equality can be further bounded as

$$
\begin{aligned}
\alpha_T \|\hat{\mathbf{g}}_T\|^2_{\hat{V}_T^{-1}} \leq& \alpha \sum_{i=1}^d \frac{\left(\sum_{j=1}^T \prod_{k=1}^{T-j} \beta_{1(T-k+1)} g_{j,i}\right)^2}{T \sum_{j=1}^T (1-\beta_{2j})\Pi_{k=1}^{T-j}\beta_{2(T-k+1)}g_{j,i}^2 + \delta} \\
\leq& \alpha \sum_{i=1}^d \frac{\left(\sum_{j=1}^T \prod_{k=1}^{T-j} \beta_{1(T-k+1)}\right)\left(\sum_{j=1}^T \prod_{k=1}^{T-j} \beta_{1(T-k+1)}g_{j,i}^2\right)}{T \sum_{j=1}^T (1-\beta_{2j})\Pi_{k=1}^{T-j}\beta_{2(T-k+1)}g_{j,i}^2 + \delta} \\
\leq& \alpha \sum_{i=1}^d \frac{\left(\sum_{j=1}^T \beta_1^{T-j}\right)\left(\sum_{j=1}^T \prod_{k=1}^{T-j} \beta_{1(T-k+1)}g_{j,i}^2\right)}{T \sum_{j=1}^T (1-\beta_{2j})\Pi_{k=1}^{T-j}\beta_{2(T-k+1)}g_{j,i}^2 + \delta} \\
\leq& \frac{\alpha}{(1-\beta_1)} \sum_{i=1}^d \frac{\sum_{j=1}^T \beta_1^{T-j}g_{j,i}^2}{T \sum_{j=1}^T (1-\beta_{2j})\Pi_{k=1}^{T-j}\beta_{2(T-k+1)}g_{j,i}^2 + \delta} \\
\overset{(9)}{\leq}& \frac{\alpha\zeta}{(1-\beta_1)} \sum_{i=1}^d \frac{\sum_{j=1}^T \beta_1^{T-j}g_{j,i}^2}{\sum_{j=1}^T g_{j,i}^2 + \zeta\delta} \leq \frac{\alpha\zeta}{(1-\beta_1)} \sum_{i=1}^d \sum_{j=1}^T \beta_1^{T-j}\frac{g_{j,i}^2}{\sum_{k=1}^j g_{k,i}^2 + \zeta\delta}
\end{aligned}
\tag{31}
$$

where the first inequality is due to $1 - \beta_{1j} \leq 1$, the second inequality follows from Cauchy-Schwarz inequality, the third inequality is due to $\beta_{1t} \leq \beta_1$. Let $r_j = g_{j,i}^2/(\sum_{k=1}^j g_{k,i}^2 + \zeta\delta)$. Using similar arguments for all time steps and summing over 1 to $T$, we have

$$
\begin{aligned}
\sum_{t=1}^T \alpha_t \|\hat{\mathbf{g}}_t\|^2_{\hat{V}_t^{-1}} \leq& \frac{\alpha\zeta}{(1-\beta_1)} \sum_{i=1}^d \sum_{t=1}^T \sum_{j=1}^t \beta_1^{t-j} r_j \\
=& \frac{\alpha\zeta}{(1-\beta_1)} \sum_{i=1}^d \sum_{j=1}^T \sum_{l=0}^{T-j} \beta_1^l r_j \\
=& \frac{\alpha\zeta}{(1-\beta_1)} \sum_{i=1}^d \sum_{j=1}^T \frac{\sum_{l=0}^{T-j} \beta_1^l g_{j,i}^2}{\sum_{k=1}^j g_{k,i}^2 + \zeta\delta} \\
\leq& \frac{\alpha\zeta}{(1-\beta_1)^2} \sum_{i=1}^d \sum_{j=1}^T \frac{g_{j,i}^2}{\sum_{k=1}^j g_{k,i}^2 + \zeta\delta} \\
\overset{(28)}{\leq}& \frac{\alpha\zeta}{(1-\beta_1)^2} \sum_{i=1}^d \log\left(\frac{\sum_{j=1}^T g_{j,i}^2}{\zeta\delta} + 1\right).
\end{aligned}
\tag{32}
$$

## D   SADAM WITH A DECAYING REGULARIZATION FACTOR

---

**Algorithm 2** SAdam with time-variant $\delta_t$ (SAdamD)

---

1: **Input:** $\{\beta_{1t}\}_{t=1}^T, \{\beta_{2t}\}_{t=1}^T, \{\delta_t\}_{t=1}^T$
2: **Initialize:** $\hat{\mathbf{g}}_0 = \mathbf{0}, \hat{V}_0 = \mathbf{0}_{d\times d}, \mathbf{x}_1 = \mathbf{0}$.
3: **for** $t = 1, \ldots, T$ **do**
4:     $\mathbf{g}_t = \nabla f_t(\mathbf{x}_t)$
5:     $\hat{\mathbf{g}}_t = \beta_{1t}\hat{\mathbf{g}}_{t-1} + (1-\beta_{1t})\mathbf{g}_t$
6:     $V_t = \beta_{2t}V_{t-1} + (1-\beta_{2t})\text{diag}(\mathbf{g}_t\mathbf{g}_t^\top)$
7:     $\hat{V}_t = V_t + \text{diag}\left(\frac{\delta_t}{t}\right)$
8:     $\mathbf{x}_{t+1} = \Pi_{\mathcal{D}}^{\hat{V}_t}\left(\mathbf{x}_t - \frac{\alpha}{t}\hat{V}_t^{-1}\hat{\mathbf{g}}_t\right)$
9: **end for**

---

In this section, we establish a generalized version of SAdam, which employs a time-variant regularization factor $\delta_{t,i}$ for each dimension $i$, instead of a fixed one for all $i \in [d]$ and $t \in [T]$ as in the original SAdam. The algorithm is referred to as SAdamD and summarized in Algorithm 2. It can be seen that

our SAdamD reduces to SC-RMSprop with time-variant $\delta_t$ when $\beta_{1t} = 0$ and $1 - \frac{1}{t} \leq \beta_{2t} \leq 1 - \frac{\gamma}{t}$. For SAdamD, we prove the following theoretical guarantee:

**Theorem 4.** *Suppose Assumptions 1 and 2 hold, and all loss functions $f_1(\cdot), \ldots, f_T(\cdot)$ are $\lambda$-strongly convex. Let $\{\delta_{t,i}\}_{t=1}^T \in (0,1]^T$ be a non-increasing sequence for all $i \in [d]$, $\beta_{1t} = \beta_1 \nu^{t-1}$ where $\beta_1 \in [0,1), \nu \in [0,1)$, and $\{\beta_{2t}\}_{t=1}^T \in [0,1]^T$ be a parameter sequence such that Conditions 3 and 4 are satisfied. Let $\alpha \geq \frac{C}{\lambda}$. The regret of SAdamD satisfies*

$$R(T) \leq \frac{D_\infty^2 \sum_{i=1}^d \delta_{1,i}}{2\alpha(1-\beta_1)} + \frac{\alpha\zeta}{(1-\beta_1)^3} \sum_{i=1}^d \log\left(\frac{1}{\zeta\delta_{T,i}} \sum_{j=1}^T g_{j,i}^2 + 1\right) + \frac{\beta_1 D_\infty^2 \left(dG_\infty^2 + \sum_{i=1}^d \delta_{1,i}\right)}{2\alpha(1-\beta_1)(\nu-1)^2}.$$

$$(33)$$

By setting $\beta_{1t} = 0$ and $1 - \frac{1}{t} \leq \beta_{2t} \leq 1 - \frac{\gamma}{t}$, we can derive the following regret bound for SC-RMSprop:

**Corollary 5.** *Suppose Assumptions 1 and 2 hold, and all loss functions $f_1(\cdot), \ldots, f_T(\cdot)$ are $\lambda$-strongly convex. Let $\{\delta_{t,i}\}_{t=1}^T \in (0,1]^T$ be a non-increasing sequence for all $i \in [d]$, and $1 - \frac{1}{t} \leq \beta_{2t} \leq 1 - \frac{\gamma}{t}$, where $\gamma \in (0,1]$. Let $\alpha \geq \frac{(2-\gamma)G_\infty^2}{\lambda}$. Then SAdamD reduces to SC-RMSprop, and the regret satisfies*

$$R(T) \leq \frac{D_\infty^2 \sum_{i=1}^d \delta_{1,i}}{2\alpha} + \frac{\alpha}{\gamma} \sum_{i=1}^d \log\left(\frac{\gamma}{\delta_{T,i}} \sum_{j=1}^T g_{j,i}^2 + 1\right).$$

$$(34)$$

Finally, we provide an instantiation of $\delta_t$ and derive the following Corollary.

**Corollary 6.** *Suppose Assumptions 1 and 2 hold, and all loss functions $f_1(\cdot), \ldots, f_T(\cdot)$ are $\lambda$-strongly convex. Let $\delta_{t,i} = \frac{\xi_2}{1+\xi_1 \sum_{j=1}^t g_{j,i}^2}$, where $\xi_2 \in (0,1]$ and $\xi_1 \geq 0$ are hypeparameters. Then we have $\delta_{t,i} \in (0,1]$ and is non-increasing $\forall i \in [d], t \in [T]$. Let $1 - \frac{1}{t} \leq \beta_{2t} \leq 1 - \frac{\gamma}{t}$, where $\gamma \in (0,1]$, and $\alpha \geq \frac{(2-\gamma)G_\infty^2}{\lambda}$. Then SAdamD reduces to SC-RMSprop, and the regret satisfies*

$$R(T) \leq \frac{dD_\infty^2 \xi_2}{2\alpha} + \frac{\alpha}{\gamma} \sum_{i=1}^d \log\left(\gamma \sum_{j=1}^T g_{j,i}^2 + \xi_2\right) + \frac{\alpha}{\gamma} \sum_{i=1}^d \log\left(\frac{\xi_1}{\xi_2} \sum_{j=1}^T g_{j,i}^2 + \frac{1}{\xi_2}\right). \quad (35)$$

## E  PROOF OF THEOREM 4

By similar arguments as in the proof of Theorem 1, we can upper bound regret as

$$R(T) \leq \underbrace{\sum_{t=1}^T \left(\frac{\|\mathbf{x}_t - \mathbf{x}_*\|_{\hat{V}_t}^2 - \|\mathbf{x}_{t+1} - \mathbf{x}_*\|_{\hat{V}_t}^2}{2\alpha_t(1-\beta_{1t})} - \frac{\lambda}{2}\|\mathbf{x}_t - \mathbf{x}_*\|^2\right)}_{P_1}$$

$$+ \underbrace{\frac{1}{2(1-\beta_1)} \sum_{t=2}^T \alpha_{t-1}\|\hat{\mathbf{g}}_{t-1}\|_{\hat{V}_{t-1}^{-1}}^2 + \frac{\sum_{t=1}^T \alpha_t \|\hat{\mathbf{g}}_t\|_{\hat{V}_t^{-1}}^2}{2(1-\beta_1)}}_{P_2} + \underbrace{\sum_{t=2}^T \frac{\beta_{1t}}{2\alpha_{t-1}(1-\beta_{1t})}\|\mathbf{x}_t - \mathbf{x}_*\|_{\hat{V}_{t-1}}^2}_{P_3}.$$

To bound $P_1$, based on (18), we have

$$P_1 \leq \sum_{t=2}^T \frac{1}{2\alpha(1-\beta_{1t})}\left(t\|\mathbf{x}_t - \mathbf{x}_*\|_{\hat{V}_t}^2 - (t-1)\|\mathbf{x}_t - \mathbf{x}_*\|_{\hat{V}_{t-1}}^2 - \lambda\alpha(1-\beta_{1t})\|\mathbf{x}_t - \mathbf{x}_*\|^2\right)$$

$$(36)$$

$$+ \left(\frac{\|\mathbf{x}_1 - \mathbf{x}_*\|_{\hat{V}_1}^2}{2\alpha_1(1-\beta_1)} - \frac{\lambda}{2}\|\mathbf{x}_1 - \mathbf{x}_*\|^2\right).$$

For the first term in (36), we have

$$t\|\mathbf{x}_t - \mathbf{x}_*\|_{\hat{V}_t}^2 - (t-1)\|\mathbf{x}_t - \mathbf{x}_*\|_{\hat{V}_{t-1}}^2 - \lambda\alpha(1-\beta_{1t})\|\mathbf{x}_t - \mathbf{x}_*\|^2$$

$$= \sum_{i=1}^d (x_{t,i} - x_{*,i})^2 (t\hat{v}_{t,i} - (t-1)\hat{v}_{t-1,i} - \lambda\alpha(1-\beta_{1t}))$$

$$= \sum_{i=1}^d (x_{t,i} - x_{*,i})^2 (tv_{t,i} - (t-1)v_{t-1,i} - \lambda\alpha(1-\beta_{1t}) + \delta_{t,i} - \delta_{t-1,i}) \tag{37}$$

$$\leq \sum_{i=1}^d (x_{t,i} - x_{*,i})^2 (\underbrace{\lambda\alpha(1-\beta_1) - \lambda\alpha(1-\beta_{1t})}_{\leq 0} + \underbrace{\delta_{t,i} - \delta_{t-1,i}}_{\leq 0})$$

$$\leq 0$$

For the second term of (36), we have

$$\left(\frac{\|\mathbf{x}_1 - \mathbf{x}_*\|_{\hat{V}_1}^2}{2\alpha_1(1-\beta_1)} - \frac{\lambda}{2}\|\mathbf{x}_1 - \mathbf{x}_*\|^2\right) = \sum_{i=1}^d (x_{1,i} - x_{*,i})^2 \left(\frac{\hat{v}_{1,i} - \lambda\alpha(1-\beta_1)}{2\alpha(1-\beta_1)}\right)$$

$$= \sum_{i=1}^d (x_{1,i} - x_{*,i})^2 \left(\frac{v_{1,i} - \lambda\alpha(1-\beta_1) + \delta_{1,i}}{2\alpha(1-\beta_1)}\right) \tag{38}$$

$$\leq \frac{D_\infty^2 \sum_{i=1}^d \delta_{1,i}}{2\alpha(1-\beta_1)}$$

where the inequality follows from Condition 3. Combining (36), (37) and (38), we have

$$P_1 \leq \frac{D_\infty^2 \sum_{i=1}^d \delta_{1,i}}{2\alpha(1-\beta_1)}. \tag{39}$$

To bound $P_2$, we first introduce the following lemma.

**Lemma 4.** *The following inequality holds*

$$\sum_{t=1}^T \alpha_t \|\hat{\mathbf{g}}_t\|_{\hat{V}_t^{-1}}^2 \leq \frac{\alpha\zeta}{(1-\beta_1)^2} \sum_{i=1}^d \log\left(\frac{1}{\zeta\delta_{T,i}} \sum_{j=1}^T g_{j,i}^2 + 1\right). \tag{40}$$

The proof of Lemma 4 can be found in Appendix F. Based on Lemma 4, we have

$$P_2 = \frac{1}{2(1-\beta_1)} \sum_{t=2}^T \alpha_{t-1} \|\hat{\mathbf{g}}_{t-1}\|_{\hat{V}_{t-1}^{-1}}^2 + \frac{\sum_{t=1}^T \alpha_t \|\hat{\mathbf{g}}_t\|_{\hat{V}_t^{-1}}^2}{2(1-\beta_1)}$$

$$\leq \frac{1}{(1-\beta_1)} \sum_{t=1}^T \alpha_t \|\hat{\mathbf{g}}_t\|_{\hat{V}_t^{-1}}^2 \tag{41}$$

$$\overset{(40)}{\leq} \frac{\alpha\zeta}{(1-\beta_1)^3} \sum_{i=1}^d \log\left(\frac{1}{\zeta\delta_{T,i}} \sum_{j=1}^T g_{j,i}^2 + 1\right).$$

Finally, we turn to upper bound $P_3$:

$$
\begin{aligned}
P_3 &\le \sum_{i=1}^{d} \sum_{t=1}^{T} \frac{\beta_{1t}}{2\alpha(1-\beta_{1t})} (x_{t,i} - x_{*,i})^2 t \hat{v}_{t,i} \\
&\le \frac{D_\infty^2}{2\alpha} \sum_{i=1}^{d} \sum_{t=1}^{T} \frac{\beta_{1t}}{1-\beta_{1t}} t(G_\infty^2 + \delta_{1,i}) \\
&\le \frac{\beta_1 D_\infty^2}{2\alpha} \sum_{i=1}^{d} \sum_{t=1}^{T} \frac{\nu^{t-1}}{1-\beta_1} t(G_\infty^2 + \delta_{1,i}) \\
&= \frac{\beta_1 D_\infty^2}{2\alpha} \sum_{i=1}^{d} \frac{(G_\infty^2 + \delta_{1,i})}{1-\beta_1} \sum_{t=0}^{T-1} \nu^t (t+1) \\
&\stackrel{(24)}{\le} \frac{\beta_1 D_\infty^2}{2\alpha} \sum_{i=1}^{d} \frac{(G_\infty^2 + \delta_{1,i})}{(1-\beta_1)(\nu-1)^2} \\
&= \frac{\beta_1 D_\infty^2 (dG_\infty^2 + \sum_{i=1}^{d} \delta_{1,i})}{2\alpha(1-\beta_1)(\nu-1)^2}.
\end{aligned}
\tag{42}
$$

We finish the proof by combining (39), (41) and (42).

## F  PROOF OF LEMMA 4

Expending the last term in the summation by using the update rule of Algorithm 2, we have

$$
\alpha_T \|\hat{\mathbf{g}}_T\|_{\hat{V}_T^{-1}}^2 = \alpha_T \sum_{i=1}^{d} \frac{\hat{g}_{T,i}^2}{v_{T,i} + \frac{\delta_{T,i}}{T}} = \alpha \sum_{i=1}^{d} \frac{\left( \sum_{j=1}^{T} (1-\beta_{1j}) \prod_{k=1}^{T-j} \beta_{1(T-k+1)} g_{j,i} \right)^2}{T \sum_{j=1}^{T} (1-\beta_{2j}) \Pi_{k=1}^{T-j} \beta_{2(T-k+1)} g_{j,i}^2 + \delta_{T,i}}.
\tag{43}
$$

The above equality can be further bounded as

$$
\begin{aligned}
\alpha_T \|\hat{\mathbf{g}}_T\|_{\hat{V}_T^{-1}}^2 &\le \alpha \sum_{i=1}^{d} \frac{\left( \sum_{j=1}^{T} \prod_{k=1}^{T-j} \beta_{1(T-k+1)} \right) \left( \sum_{j=1}^{T} \prod_{k=1}^{T-j} \beta_{1(T-k+1)} g_{j,i}^2 \right)}{T \sum_{j=1}^{T} (1-\beta_{2j}) \Pi_{k=1}^{T-j} \beta_{2(T-k+1)} g_{j,i}^2 + \delta_{T,i}} \\
&\le \alpha \sum_{i=1}^{d} \frac{\left( \sum_{j=1}^{T} \beta_1^{T-j} \right) \left( \sum_{j=1}^{T} \prod_{k=1}^{T-j} \beta_{1(T-k+1)} g_{j,i}^2 \right)}{T \sum_{j=1}^{T} (1-\beta_{2j}) \Pi_{k=1}^{T-j} \beta_{2(T-k+1)} g_{j,i}^2 + \delta_{T,i}} \\
&\le \frac{\alpha}{(1-\beta_1)} \sum_{i=1}^{d} \frac{\sum_{j=1}^{T} \beta_1^{T-j} g_{j,i}^2}{T \sum_{j=1}^{T} (1-\beta_{2j}) \Pi_{k=1}^{T-j} \beta_{2(T-k+1)} g_{j,i}^2 + \delta_{T,i}} \\
&\stackrel{(9)}{\le} \frac{\alpha\zeta}{(1-\beta_1)} \sum_{i=1}^{d} \frac{\sum_{j=1}^{T} \beta_1^{T-j} g_{j,i}^2}{\sum_{j=1}^{T} g_{j,i}^2 + \zeta\delta_{T,i}} \\
&\le \frac{\alpha\zeta}{(1-\beta_1)} \sum_{i=1}^{d} \sum_{j=1}^{T} \beta_1^{T-j} \frac{g_{j,i}^2}{\sum_{k=1}^{j} g_{k,i}^2 + \zeta\delta_{T,i}}.
\end{aligned}
\tag{44}
$$

The first inequality follows from Cauchy-Schwarz inequality and $1 - \beta_{1t} \le 1$, and the second inequality is due to $\beta_{1t} \le \beta_1$. Let $r_j = \frac{g_{j,i}^2}{\sum_{k=1}^{j} g_{k,i}^2 + \zeta\delta_{T,i}}$. By using similar arguments as in (32), we

have

$$
\begin{aligned}
\sum_{t=1}^{T} \alpha_t \|\hat{\mathbf{g}}_t\|_{\hat{V}_t^{-1}}^2 &\leq \frac{\alpha\zeta}{(1-\beta_1)} \sum_{i=1}^{d} \sum_{t=1}^{T} \sum_{j=1}^{t} \beta_1^{T-j} \frac{g_{j,i}^2}{\sum_{k=1}^{j} g_{k,i}^2 + \zeta\delta_{t,i}} \\
&\leq \frac{\alpha\zeta}{(1-\beta_1)} \sum_{i=1}^{d} \sum_{t=1}^{T} \sum_{j=1}^{t} \beta_1^{T-j} \frac{g_{j,i}^2}{\sum_{k=1}^{j} g_{k,i}^2 + \zeta\delta_{T,i}} \\
&= \frac{\alpha\zeta}{(1-\beta_1)} \sum_{i=1}^{d} \sum_{t=1}^{T} \sum_{j=1}^{t} r_j \\
&\leq \frac{\alpha\zeta}{(1-\beta_1)} \sum_{i=1}^{d} \sum_{j=1}^{T} \frac{\sum_{l=0}^{T-j} \beta_1^l g_{j,i}^2}{\sum_{k=1}^{j} g_{k,i}^2 + \zeta\delta_{T,i}} \\
&\leq \frac{\alpha\zeta}{(1-\beta_1)^2} \sum_{i=1}^{d} \sum_{j=1}^{T} \frac{g_{j,i}^2}{\sum_{k=1}^{j} g_{k,i}^2 + \zeta\delta_{T,i}} \\
&\overset{(28)}{\leq} \frac{\alpha\zeta}{(1-\beta_1)^2} \sum_{i=1}^{d} \log\left(\frac{1}{\zeta\delta_{T,i}} \sum_{j=1}^{T} g_{j,i}^2 + 1\right).
\end{aligned}
\tag{45}
$$

## G    ON THE CONVERGENCE OF AMSGRAD

In this section, we firstly provide the AMSgrad algorithm and its theoretical guarantees (Reddi et al., 2018), then state a theoretical flaw in their analysis revealed by Tran et al. (2019), and finally propose a simple solution to fix this problem.

The AMSgrad algorithm developed in Reddi et al. (2018) is summarized in Algorithm 3.

---

**Algorithm 3** AMSgrad

1: **Input:** $\{\beta_{1t}\}_{t=1}^T, \beta_2$
2: **Initialize:** $\hat{\mathbf{g}}_0 = \mathbf{0}$, $\hat{V}_0 = \mathbf{0}_{d\times d}$, $\mathbf{x}_1 = \mathbf{0}$.
3: **for** $t = 1, \ldots, T$ **do**
4:     $\mathbf{g}_t = \nabla f_t(\mathbf{x}_t)$
5:     $\hat{\mathbf{g}}_t = \beta_{1t}\hat{\mathbf{g}}_{t-1} + (1-\beta_{1t})\mathbf{g}_t$
6:     $V_t = \beta_2 V_{t-1} + (1-\beta_2)\mathrm{diag}(\mathbf{g}_t\mathbf{g}_t^\top)$
7:     $\hat{V}_t = \max\{V_t, \hat{V}_{t-1}\}$
8:     $\mathbf{x}_{t+1} = \Pi_{\mathcal{D}}^{\sqrt{\hat{V}_t}}\left(\mathbf{x}_t - \alpha_t\hat{V}_t^{-1/2}\hat{\mathbf{g}}_t\right)$, where $\alpha_t = \frac{\alpha}{\sqrt{t}}$
9: **end for**

---

For AMSgrad, Reddi et al. (2018) provide the following regret bound.

**Theorem 7** (Theorem 4 in Reddi et al. (2018), problematic)**.** *Suppose Assumptions 1 and 2 hold, and all loss functions $f_1(\cdot), \ldots, f_T(\cdot)$ are convex. Let $\delta > 0$, $\beta_1 > 0$, $\beta_{1t} \leq \beta_1$, and $\gamma = \frac{\beta_1}{\sqrt{\beta_2}} \leq 1$. The regret of AMSgrad satisfies*

$$
\begin{aligned}
R(T) \leq &\frac{D_\infty^2 \sqrt{T}}{\alpha(1-\beta_1)} \sum_{i=1}^{d} \hat{v}_{T,i}^{1/2} + \frac{D_\infty^2}{2(1-\beta_1)} \sum_{t=1}^{T} \sum_{i=1}^{d} \frac{\beta_{1t}\hat{v}_{t,i}^{1/2}}{\alpha_t} \\
&+ \frac{\alpha\sqrt{1+\log T}}{(1-\beta_1)^2(1-\gamma)\sqrt{(1-\beta_2)}} \sum_{i=1}^{d} \|g_{1:T,i}\|.
\end{aligned}
\tag{46}
$$

Recently, Tran et al. (2019) point out a mistake in the proof of Theorem 7. Specifically, in Reddi et al. (2018), the following inequality is utilized (Proof of Lemma 2, Page 18):

$$
\sum_{t=1}^{T} \left[ \frac{1}{2\alpha_t(1-\beta_{1t})} \left[ \|\hat{V}_t^{1/4}(\mathbf{x}_t - \mathbf{x}_*)\|^2 - \|\hat{V}_t^{1/4}(\mathbf{x}_{t+1} - \mathbf{x}_*)\|^2 \right] \right.
$$

$$
\left. + \frac{\beta_{1t}}{2\alpha_t(1-\beta_{1t})} \|\hat{V}_t^{1/4}(\mathbf{x}_t - \mathbf{x}_*)\|^2 \right] + \frac{\alpha\sqrt{1+\log T}}{(1-\beta_1)^2(1-\gamma)\sqrt{(1-\beta_2)}} \sum_{i=1}^{d} \|g_{1:T,i}\|
$$

$$
\leq \frac{1}{2\alpha_1(1-\beta_1)} \|\hat{V}_t^{1/4}(\mathbf{x}_1 - \mathbf{x}_*)\|^2 + \frac{1}{2(1-\beta_1)} \sum_{t=2}^{T} \left[ \frac{\|\hat{V}_t^{1/4}(\mathbf{x}_t - \mathbf{x}_*)\|^2}{\alpha_t} - \frac{\|\hat{V}_{t-1}^{1/4}(\mathbf{x}_t - \mathbf{x}_*)\|^2}{\alpha_{t-1}} \right]
$$

$$
+ \sum_{t=1}^{T} \left[ \frac{\beta_{1t}}{2\alpha_t(1-\beta_1)} \|\hat{V}_t^{1/4}(\mathbf{x}_t - \mathbf{x}_*)\|^2 \right] + \frac{\alpha\sqrt{1+\log T}}{(1-\beta_1)^2(1-\gamma)(\sqrt{1-\beta_2})} \sum_{i=1}^{d} \|g_{1:T,i}\|
$$

$$(47)$$

which, however, may not hold. To see this, we note that essentially (47) uses

$$
\sum_{t=1}^{T} \frac{1}{2\alpha_t(1-\beta_{1t})} \left[ \|\hat{V}_t^{1/4}(\mathbf{x}_t - \mathbf{x}_*)\|^2 - \|\hat{V}_t^{1/4}(\mathbf{x}_{t+1} - \mathbf{x}_*)\|^2 \right]
$$

$$
\leq \frac{1}{(1-\beta_1)} \sum_{t=1}^{T} \frac{1}{2\alpha_t} \left[ \|\hat{V}_t^{1/4}(\mathbf{x}_t - \mathbf{x}_*)\|^2 - \|\hat{V}_t^{1/4}(\mathbf{x}_{t+1} - \mathbf{x}_*)\|^2 \right]
$$

$$(48)$$

which holds only if $\beta_{1t} \leq \beta_1$ and $\|\hat{V}_t^{1/4}(\mathbf{x}_t - \mathbf{x}_*)\|^2 - \|\hat{V}_t^{1/4}(\mathbf{x}_{t+1} - \mathbf{x}_*)\|^2$ is non-negative. However, as empirically shown by Tran et al. (2019), the letter requirement can be violated in some counterexamples. Note that similar problems exist in many recent proposed Adam variants. To address this issue, Tran et al. (2019) establish a new convergence proof of AMSGrad, which indicates an $O(d\sqrt{T})$ *data-independent* regret bound. Moreover, as an alternative, they also propose a variant of AMSgrad, called AdamX, which alters the stricture of AMSgrad to force the inequality been satisfied. For AdamX, they also give a new theoretical analysis and an $O(d\sqrt{T})$ *data-independent* regret bound.

In this paper, we find out that the above problem can be solved by simply configuring $\beta_{1t}$ of AMSgrad in a non-increasing manner, i.e., $\forall t \geq 2, \beta_{1t} \leq \beta_{1(t-1)}$. Specifically, when $\beta_{1t}$ is non-increasing, we can rewrite (47) as

$$
\sum_{t=1}^{T} \left[ \frac{1}{2\alpha_t(1-\beta_{1t})} \left[ \|\hat{V}_t^{1/4}(\mathbf{x}_t - \mathbf{x}_*)\|^2 - \|\hat{V}_t^{1/4}(\mathbf{x}_{t+1} - \mathbf{x}_*)\|^2 \right] \right.
$$

$$
\left. + \frac{\beta_{1t}}{2\alpha_t(1-\beta_{1t})} \|\hat{V}_t^{1/4}(\mathbf{x}_t - \mathbf{x}_*)\|^2 \right] + \frac{\alpha\sqrt{1+\log T}}{(1-\beta_1)^2(1-\gamma)\sqrt{(1-\beta_2)}} \sum_{i=1}^{T} \|g_{1:T,i}\|
$$

$$
\leq \frac{1}{2\alpha_1(1-\beta_1)} \|\hat{V}_1^{1/4}(\mathbf{x}_1 - \mathbf{x}_*)\|^2 + \frac{1}{2} \sum_{t=2}^{T} \left[ \frac{\|\hat{V}_t^{1/4}(\mathbf{x}_t - \mathbf{x}_*)\|^2}{\alpha_t(1-\beta_{1t})} - \frac{\|\hat{V}_{t-1}^{1/4}(\mathbf{x}_t - \mathbf{x}_*)\|^2}{\alpha_{t-1}(1-\beta_{1(t-1)})} \right]
$$

$$
+ \sum_{t=1}^{T} \left[ \frac{\beta_{1t}}{2\alpha_t(1-\beta_1)} \|\hat{V}_t^{1/4}(\mathbf{x}_t - \mathbf{x}_*)\|^2 \right] + \frac{\alpha\sqrt{1+\log T}}{(1-\beta_1)^2(1-\gamma)(\sqrt{1-\beta_2})} \sum_{i=1}^{d} \|g_{1:T,i}\|
$$

$$
\leq \frac{1}{2\alpha_1(1-\beta_1)} \|\hat{V}_1^{1/4}(\mathbf{x}_1 - \mathbf{x}_*)\|^2 + \frac{1}{2} \sum_{t=2}^{T} \left[ \frac{\|\hat{V}_t^{1/4}(\mathbf{x}_t - \mathbf{x}_*)\|^2}{\alpha_t(1-\beta_{1t})} - \frac{\|\hat{V}_{t-1}^{1/4}(\mathbf{x}_t - \mathbf{x}_*)\|^2}{\alpha_{t-1}(1-\beta_{1t})} \right]
$$

$$
+ \sum_{t=1}^{T} \left[ \frac{\beta_{1t}}{2\alpha_t(1-\beta_1)} \|\hat{V}_t^{1/4}(\mathbf{x}_t - \mathbf{x}_*)\|^2 \right] + \frac{\alpha\sqrt{1+\log T}}{(1-\beta_1)^2(1-\gamma)(\sqrt{1-\beta_2})} \sum_{i=1}^{d} \|g_{1:T,i}\|
$$

$$\leq \frac{1}{2\alpha_1(1-\beta_1)}\|\hat{V}_1^{1/4}(\mathbf{x}_1-\mathbf{x}_*)\|^2 + \frac{1}{2(1-\beta_1)}\sum_{t=2}^{T}\left[\frac{\|\hat{V}_t^{1/4}(\mathbf{x}_t-\mathbf{x}_*)\|^2}{\alpha_t} - \frac{\|\hat{V}_{t-1}^{1/4}(\mathbf{x}_t-\mathbf{x}_*)\|^2}{\alpha_{t-1}}\right]$$

$$+ \sum_{t=1}^{T}\left[\frac{\beta_{1t}}{2\alpha_t(1-\beta_1)}\|\hat{V}_t^{1/4}(\mathbf{x}_t-\mathbf{x}_*)\|^2\right] + \frac{\alpha\sqrt{1+\log T}}{(1-\beta_1)^2(1-\gamma)(\sqrt{1-\beta_2})}\sum_{i=1}^{d}\|g_{1:T,i}\|$$

where the second inequality is derived from $\beta_{1t} \leq \beta_{1(t-1)}$, and the last inequality is due to the fact that $\beta_{1t} \leq \beta_1$ and $\left[\frac{\|\hat{V}_t^{1/4}(\mathbf{x}_t-\mathbf{x}_*)\|^2}{\alpha_t} - \frac{\|\hat{V}_{t-1}^{1/4}(\mathbf{x}_t-\mathbf{x}_*)\|^2}{\alpha_{t-1}}\right]$ is non-negative. In this way, the proof of AMSgrad can proceed, and the algorithm structure as well as the conclusion in Theorem 7 remain unchanged.

To summarize, we restate Theorem 7 as follows.

**Theorem 8** (Fixed theoretical guarantee of AMSgrad). *Suppose Assumptions 1 and 2 hold, and all loss functions $f_1(\cdot), \ldots, f_T(\cdot)$ are convex. Let $\delta > 0$, $\beta_1 > 0$, $\beta_{1t} \leq \beta_{1(t-1)}$ where $t \geq 2$, $\beta_{11} = \beta_1$, and $\gamma = \frac{\beta_1}{\sqrt{\beta_2}} \leq 1$. The regret of AMSgrad satisfies*

$$R(T) \leq \frac{D_\infty^2 \sqrt{T}}{\alpha(1-\beta_1)}\sum_{i=1}^{d}\hat{v}_{T,i}^{1/2} + \frac{D_\infty^2}{2(1-\beta_1)}\sum_{t=1}^{T}\sum_{i=1}^{d}\frac{\beta_{1t}\hat{v}_{t,i}^{1/2}}{\alpha_t}$$

$$+ \frac{\alpha\sqrt{1+\log T}}{(1-\beta_1)^2(1-\gamma)\sqrt{(1-\beta_2)}}\sum_{i=1}^{d}\|g_{1:T,i}\|. \tag{49}$$

## H    EXPERIMENTS ON RESNET18

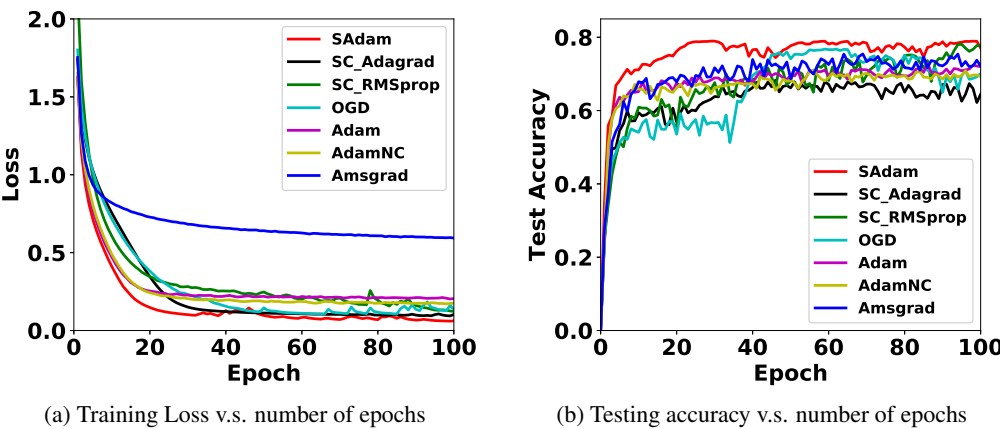

(a) Training Loss v.s. number of epochs

(b) Testing accuracy v.s. number of epochs

Fig. 4: experimental results of ResNet18 on CIFAR10 dataset

In this section, we conduct the image classification task on CIFAR10 dataset using ResNet18 (He et al., 2016). The parameter configuration of algorithms is the same as that in Section 4. We repeat each experiment 10 times and take their average. The training loss v.s. epoch is shown in Fig. 4a, and the testing accuracy v.s. epoch is shown in Fig. 4b. As can be seen, our proposed SAdam outperforms other algorithms.

