# OpenReview forum: "SAdam: A Variant of Adam for Strongly Convex Functions"
_ICLR.cc/2020/Conference — Accept (Poster)_

### Official Review · AnonReviewer3 · 2019-10-21
**Official Blind Review #3**

**Rating:** 8

**Review:**

In the setting of online convex optimization, this paper investigates the question of whether adaptive gradient methods can achieve “data dependent” logarithmic regret bounds when the class of loss functions is strongly convex. To this end, the authors propose a variant of Adam - called SAdam - which indeed satisfies such a desired bound. Importantly, SAdam is an extension of SC-RMSprop (a variant of RMSprop) for which a “data independent” logarithmic bound was found. Experiments on optimizing strongly convex functions and training deep networks show that SAdam outperforms other adaptive gradient methods (and SGD).

The paper is very well-written, well-motivated and well-positioned with respect to related work. The regret analysis of SAdam is conceptually simple and elegant. The experimental protocol is well-detailed, and the results look promising. In a nutshell, this is an excellent piece of work.

I have just a minor comment. In the experiments, SAdam was tested using $\beta_1 = 0.9$ and $\beta_{2t} = 1 - \frac{0.9}{t}$. Since Corollary 2 covers a wide range of admissible values for these parameters, it would be interesting to report (for example in Appendix) a sensitivity analysis of SAdam, using different choices of $\beta_1$ and $\beta_{2t}$.


**Experience Assessment:**

I have published one or two papers in this area.

**Review Assessment: Checking Correctness Of Derivations And Theory:**

I carefully checked the derivations and theory.

**Review Assessment: Checking Correctness Of Experiments:**

I assessed the sensibility of the experiments.

**Review Assessment: Thoroughness In Paper Reading:**

I read the paper thoroughly.

---

> ### Author Response · Authors · 2019-11-10
> **Response to Review #3**
>
> Thanks for your comments!
>
> Q1: It would be interesting to report (for example in Appendix) a sensitivity analysis of SAdam, using different choices of \beta_1 and \beta_{2t}.
> A1: Thanks for your constructive suggestion and we will provide experimental results about the sensitivity with respect to \beta_1 and \beta_{2t} in the revised version. From our experience, SAdam performs well in a wide range of hyper-parameter choices.

---

### Official Review · AnonReviewer2 · 2019-10-22
**Official Blind Review #2**

**Rating:** 6

**Review:**

In this paper, the authors propose a variant of Adam, named as SAdam, and establish a data-dependent O(log T) regret bound. The key idea is using a faster decaying yet under controlled step size to exploit strong convexity. Some experiments are carried out to demonstrate the effectiveness of the proposed algorithm. The idea seems interesting, the writing is well-written, and the analysis seems correct (I did not fully check all steps, but the key steps seems ok to me).

Probs:
1. The proposed SAdam is an effective variant of Adam designed for strongly convex functions. The algorithm is a natural extension of Adam, and SC-RMSprop could be regarded as a special case.
2. The authors establish a data-dependent O(log T) regret bound for SAdam, and as a byproduct, they present the first data-dependent logarithmic regret for SC-RMSprop. The authors also fix a small bug in the analysis of AMSgrad. The theoretical result is the key technical contribution of this paper.
3. The experimental results shows that Aadam can be used to minimize strongly convex functions, as well as neural networks, which is believed to be non-convex.

Cons:
1. As the authors mentioned in Remark 2, the main limitation of their analysis is that the role of the first-order momentum is unclear. Although the first-order momentum can accelerate the convergence in practice, proving this in theory remains an open problem. Is there some contribution on this aspect?

2. The 4-layer CNN in Section 4.2 is a bit small. It would be better if the authors test their algorithm on larger and more popular neural networks.

In summary, this paper contributes the theoretical studies of ADAM-type algorithm, although the algorithm is somehow incremental. To me, a bit surprising result is that the step size originally designed for strongly convex functions also works well for training CNN.


**Experience Assessment:**

I have published one or two papers in this area.

**Review Assessment: Checking Correctness Of Derivations And Theory:**

I assessed the sensibility of the derivations and theory.

**Review Assessment: Checking Correctness Of Experiments:**

I carefully checked the experiments.

**Review Assessment: Thoroughness In Paper Reading:**

I read the paper at least twice and used my best judgement in assessing the paper.

---

> ### Author Response · Authors · 2019-11-10
> **Response to Review #2**
>
> Thanks for the comments!
>
> Q1: “The role of the first-order momentum is unclear…is there some contribution on this aspect?”
> A1: While our paper is the first to show that algorithms equipped with first-order momentum can achieve logarithmic regret bound for strongly convex functions, it is still an open problem to explicitly analysis the influence of this procedure. We note that all the regret bounds of Adam-like algorithms (e.g., Reddi et al., 2018; Chen et al., 2018a) suffer this limitation, and the advantage of first-order momentum is mainly proved by empirical studies. The difficulty is caused by the fact that the regret bound is data-dependent. Specifically, our regret bounds depend on the cumulation of all gradients g_t, and each g_t would be affected by the first-order momentum. We would like to analyze the influence of first-order momentum theoretically in the future.
>
>
> Q2: The 4-layer CNN in Section 4.2 is a bit small. It would be better if the authors test their algorithm on larger and more popular neural networks.
> A2: Thanks for the suggestion. We have applied our algorithm to the training of ResNet18 (He et al., 2016) in Appendix H. We are sorry that we forget to mention this clearly in the main paper, and will provide more experiments in the full version.

---

> > ### Comment · AnonReviewer2 · 2019-11-13
> > **saw the resp**
> >
> >  I saw the additional experiments on ResNet18 in Appendix H. So, the paper performs experiments on training one strongly convex function ($\ell_2$-regularized softmax regression), and two neural networks (4-layer CNN and ResNet18). I am satisfied with the experiments now and feel they are sufficient for a conference paper.
> >
> > Regarding the theoretical analysis of the first-order momentum, I understand this is a challenging problem, and encourage the authors to investigate it.

---

### Official Review · AnonReviewer1 · 2019-10-23
**Official Blind Review #1**

**Rating:** 3

**Review:**

This paper studies Adam and proves that under strong convexity assumption, it obtains the improved regret bound $O(log(T))$. The regret bound is data-dependent, thus as a side-effect it also improves previous known result for strongly convex RMSProp (SC-RMSProp).

The paper is clear and well-written and I also think that theoretical results are correct and new. However, I have some concerns on the possible impacts of the results especially in the context of ICLR:

- First of all, the assumption to show improved regret is strong convexity of all functions $f_t$. However, this is very restrictive and much stronger than the assumption that the sum of functions is strongly convex. In addition, from what I see, in the proof of Theorem 1, the authors use strong convexity with the $x^\star$, so they can maybe replace global strong convexity to strong convexity restricted to the path towards the solution. A reference where these restricted strong convexity type assumptions are studied:

Necoara, Nesterov, Glineur, “Linear convergence of first order methods for non-strongly convex optimization”, Math. Prog. 2019.

- To show improved regret, consistent with previous work SC-Adagrad and SC-RMSProp, the authors modify the algorithm to use $V_t^{-1}$ in page 3 in the step size, instead of $V_t^{-1/2}$ of regular Adam. It is easy to see that this is to make sure step size has a faster decrease, which is needed also to show standard SGD gets $1/k$ rate for strongly convex problems. However, given that one might not now if the problem has strong convexity (there might exist cases where this property only exists locally), it is not clear if one should apply Adam or SAdam.

- Another remark related to the previous one is the following. Standard SGD uses step size $\alpha_0/\sqrt{k}$ for convex optimization without strong convexity and $\alpha_0/k$ for strongly convex optimization. If one uses $alpha_0/k $for convex optimization without strong convexity, one gets a very bad rate $1/log(k)$ and very bad practical performance. So, given SAdam gives step sizes suited for strongly convex optimization (similar to SGD for strongly convex optimization), I would expect SAdam's step sizes to be not very suitable when there is no strong convexity.

- An additional point is that the step size of SAdam depends on the global strong convexity parameter $\lambda$ which further restricts the applicability of the method. For the theoretical results to hold, the step size should be set according to $\lambda$, and when the step size is not selected that way, one loses the fast convergence rate.

- In the experiments, the authors show the performance of SAdam for neural network training and related to my previous remarks, I have the following concerns. First of all, how do the authors pick step sizes now since it depends on strong convexity constant as in eq. (7). In addition, given that neural networks are certainly non-strongly convex, I would expect that the fast decreasing step size caused by using $V_t^{-1}$ might also hurt the performance considerably, which happens as I discussed above even for convex but non-strongly convex losses. I would suspect that much worse effects can be seen for non-convex optimization. Of course, the authors can argue that if the loss landscape of neural network has some local strong convexity parameters, SAdam would adapt and get faster convergence. But unfortunately, I would not agree with such a statement, because the analysis is not made to adapt to local strong convexity and a dependence to strong convexity constant is present due to eq. (7), so if one does not know the constant, the theoretical guarantees would not apply. In addition, the provided experiments for neural network training is not extensive enough to convince practitioners to use SAdam instead of Adam which has been used for years.

Overall, I think that it is interesting to see that a variant of Adam can be shown to obtain improved regret under strong convexity, I find the assumptions strong and the impact for neural network training, therefore for ICLR, quite questionable.

**Experience Assessment:**

I have published one or two papers in this area.

**Review Assessment: Checking Correctness Of Derivations And Theory:**

I assessed the sensibility of the derivations and theory.

**Review Assessment: Checking Correctness Of Experiments:**

I assessed the sensibility of the experiments.

**Review Assessment: Thoroughness In Paper Reading:**

I read the paper thoroughly.

---

> ### Author Response · Authors · 2019-11-10
> **Response to Review #1**
>
> Thanks for the comments!
>
> Q1: “In the proof of Theorem 1, the authors use strong convexity with the x_*, so they can maybe replace global strong convexity to strong convexity restricted to the path towards the solution.”
> A1: Thanks for the suggestion. We agree that it could be possible to replace the global strong convexity to restricted strong convexity. Then, we probably need to study the stochastic setting, and bound the excess risk instead of the regret. It is difficult to exploit restricted strong convexity in the analysis of regret of OCO. Because in the online setting, the optimal solution of each f_t is different and may not be x_*. Therefore, if we replace global strong convexity of each f_t by restricted strong convexity (with respect to its own optimal solution), the inequality about x_* (eq.(13)) can not be obtained, unless all f_t share the same optimal solution. We will study the restricted strong convexity as a future work.
>
> Q2: “Given that one might not know if the problem has strong convexity, … it is not clear if one should apply Adam or SAdam…I would expect SAdam's step sizes to be not very suitable when there is no strong convexity…the step size of SAdam depends on the global strong convexity parameter lambda.”
> A2: First, we would like to emphasize that optimization under lambda-strong convexity is a classic problem which has been widely studied in both OCO and stochastic optimization (e.g., Hazan et al., 2007, Hazan & Kyle, 2014, Mukkamala & Hein, 2017, Chen et al. 2018). The problem is important by its own right.
> Second, if the type of loss functions or the value of lambda is unknown to the learner, it is possible to combine the theoretical guarantees of Adam and SAdam by applying the universal algorithm framework (van Erven et al., 2016, Wang et al., 2019). The key idea is to simultaneously run multiple copies of each algorithm with different learning rates in every round, and adaptively learn the best one on the fly. In this way, the algorithm can handle both convex and strongly convex functions, and does not need to know any prior knowledge of lambda. It is an interesting problem and will be investigated in the future.
>
> Q3: How do the authors pick step sizes now since it depends on strong convexity constant as in eq. (7).
> A3: Following previous work (Mukkamala & Hein, 2017), for all optimization algorithms, we pick the step sizes in the set {10^{-1},10^{-2},10^{-3} , 10^{-4}} and report the best results.
>
> Q4: “Given that neural networks are certainly non-strongly convex…I would suspect that much worse effects can be seen for non-convex optimization.”
> A4: We agree that currently there still exists a gap between the theoretical analysis of SAdam and its applications to training networks. However, we note that, initially, most of the popular algorithms such as Adagrad, Adam, AMSgrad and SC-RMSprop, are analyzed under the convex assumption or strongly convex assumption. Although these assumptions are violated in training networks, these algorithms have exhibited outstanding results in the experiments. Moreover, the analysis in convex setting lays the foundations of many follow-up works that investigate the non-convex problems (e.g., Basu et al., 2018, Chen et al., 2019, Staib et al., 2019). In this paper, we prove that our proposed SAdam is able to attain tighter regret bounds under strongly convex condition, and empirically show that it achieves better performance for training some networks. We believe our results are meaningful and could inspire the analysis of Adam-type algorithms under non-convex settings.
>
>
> T. van Erven, and W. M. Koolen. Metagrad: Multiple learning rates in online learning. In NIPS, pages 3666–3674, 2016.
> G. Wang, S. Lu, and L. Zhang. Adaptivity and optimality: A universal algorithm for online convex optimization. In UAI, 2019.
> M. Staib, S. J. Reddi, S. Kale, S. Kumar, & S. Sra. Escaping saddle points with adaptive gradient methods. arXiv preprint arXiv:1901.09149, 2019.

---

> > ### Comment · AnonReviewer1 · 2019-11-15
> > **Response to authors**
> >
> > I thank the authors for their response and I provide further remarks below.
> >
> > A1:
> >
> > Thank you for the explanation. I see that global strong convexity of $f_t$ is a common assumption for online convex optimization, used for OGD, Adagrad and RMSProp in the literature. But indeed, it would be nice to have a weaker assumption for the stochastic optimization setting.
> >
> > A2:
> >
> > It is true that it might be possible to combine the guarantees in this paper with Metagrad-type framework, and the results of this paper would be important in this case. Authors can use such an approach instead of tuning the algorithms etc, but I see that line of work to be independent for evaluating this paper.
> >
> > A3:
> >
> > Thank you for the experimental clarification.
> >
> > A4:
> >
> > I disagree on this point. It is true that OGD, Adagrad and RMSProp are analyzed under convex and strongly convex settings. However, the idea to improve the regret bound under strong convexity is essentially the same in all of them. We want to have a faster decreasing step size when there is strong convexity. It is clear how to achieve this in the non-adaptive case of OGD, one takes a step size $\alpha_0/k$ where $\alpha_0$ depends on strong convexity constant, instead of standard $\alpha_0/\sqrt{k}$. For the adaptive case, the idea goes back to a technical report of Duchi, Hazan, Singer from 2010 (please see Section 5 and discussions therein):
> >
> > J. Duchi, E. Hazan, and Y. Singer. Adaptive subgradient methods for online learning and stochastic optimization. Technical Report 2010-24, UC Berkeley Electrical Engineering and Computer Science
> > https://www2.eecs.berkeley.edu/Pubs/TechRpts/2010/EECS-2010-24.pdf
> >
> > The idea for getting a smaller regret bound is therefore common for Adagrad, RMSProp and in this paper, for Adam. Square root is removed from the term $\hat{V}$ to have this faster decrease. Therefore, it is not surprising that the same modification improves the regret of Adam for strongly convex functions, after Adagrad and RMSProp.
> >
> > Second, I do not agree that the analysis of this paper will help on showing guarantees for nonconvex optimization with Adam. It is true that analyses for convex case, might give insights on the nonconvex studies. However, in the *strongly convex* case, the manipulation needed for improving regret for these methods is very specific to structure (i.e. strong convexity) and well-known. One can exploit the additional quadratic term coming from strong convexity to be able to use a faster decreasing step size, leading to smaller regret. But I am very doubtful that this idea will be useful for showing convergence for nonconvex optimization, when such a structure is not present.
> >
> > Lastly, I see that this method can be useful in practice for neural networks, so it can be a nice heuristic that people might try when training neural networks. However, I think that this part is orthogonal to the theoretical contribution of the paper, which is to have smaller regret for strongly convex functions.
> >
> > Finally, I think that this is indeed a solid work, however I do not think that the theoretical results of this paper are very insightful for ICLR community on neural network training or nonconvex optimization. Practical results might be of independent interest as a heuristic, but I do not find those enough for accepting the paper.

---

> > > ### Author Response · Authors · 2019-11-15
> > > **Response to Review #1**
> > >
> > > Thanks for your response!
> > >
> > > Q1: Square root is removed from the term to have this faster decrease. Therefore, it is not surprising that the same modification improves the regret of Adam for strongly convex functions, after Adagrad and RMSProp.
> > >
> > > A1: We agree that the idea of removing the square root from $V$ is inspired by previous work. However, we note that the analysis of SAdam is more complicated since it involves the first-order momentum as well as a more general $\beta_{2t}$ (compared to SC-RMSprop). Moreover, the regret bound for SC-RMSprop provided by Mukkamala & Hein (2017) is data-independent, while we derived the first data-dependent regret bound for SC-RMSprop.
> > >
> > > Q2: in the *strongly convex* case, the manipulation needed for improving regret for these methods is very specific to structure (i.e. strong convexity) and well-known. One can exploit the additional quadratic term coming from strong convexity to be able to use a faster decreasing step size, leading to smaller regret. But I am very doubtful that this idea will be useful for showing convergence for nonconvex optimization, when such a structure is not present.
> > >
> > > A2: We agree that global strong convexity does not hold in the non-convex setting. However, many real-world applications, including tensor decomposition (Ge et al., 2015), matrix sensing (Bhojanapalli et al., 2016), and over-parameterized neural networks (Du et al., 2019), exhibit strong local geometric properties similar to strong convexity in the global setting, and exploiting these properties may lead to much faster convergence to local (or global) minima. Therefore, we believe that our work is meaningful and will inspire the analysis of Adam-type algorithms under many non-convex settings that enjoy such strong convexity-like properties.
> > > To give a simple example, we consider the strict saddle condition defined in Ge et al. (2015), which assumes that the loss function is strongly convex in a region close to local minimum. In their paper (Remark 7), the authors firstly use a noised version of SGD to output a point that is close to a local minimum, then employ standard SGD with step size 1/t to ensure that the algorithm can converge. The analysis of both procedures depends on the local strong convexity. A naïve replacement of the latter algorithm with SAdam may lead to faster data-dependent theoretical guarantees, and it is also interesting to investigate whether a noised version of SAdam can find a point close to a local minimum faster under this condition.
> > >
> > >
> > >
> > > R. Ge, F. Huang, C. Jin, & Y. Yuan. Escaping from saddle points—online stochastic gradient for tensor decomposition. In COLT, 2015.
> > > S. Bhojanapalli, B. Neyshabur, & N. Srebro. Global optimality of local search for low rank matrix recovery. In NIPS, 2016.
> > > S. Du, X. Zhai, B. Poczos, & A. Singh. Gradient Descent Provably Optimizes Over-parameterized Neural Networks. In ICLR, 2019.

---

> > > > ### Comment · AnonReviewer1 · 2019-11-15
> > > > **Response to authors**
> > > >
> > > > I thank the authors for their response. I provide further remarks:
> > > >
> > > > A1: I agree that analysis of Adam is more involved compared to RMSProp and I appreciate the authors' efforts in the paper and repeat that I think it is solid. However, given that ICLR is a specialized conference (compared to ICML or Neurips), assuming a very strong structure, which does not hold in many applications that ICLR community is interested in, causes me to justify the suitability of the paper for the conference.
> > > >
> > > > A2: I agree with the authors that their proposed ideas are interesting and worth studying. However, it is not easy to argue that these ideas are obtainable from the results of this paper. For example, in Ge et al. (2015), assumption of strong convexity is on the function $f(x) = E[\phi(x)]$, whereas this paper's assumption, translated to the language of Ge et al. (2015) is on the component functions $\phi(x)$ which is a much stronger assumption than strong convexity of $f(x)$. But as the authors pointed out earlier, it might be possible to relax the assumption, when specialized to stochastic optimization, but this is a future study. In addition, in Remark 7 in Ge et al. (2015), the authors use the step size $1/t$ to obtain faster convergence. I agree that a similar analysis can be carried out for SAdam (or a noised version), but this is also a future study.
> > > >
> > > > To be clear on my previous comments on nonconvex optimization, the current research trend that is also cited by the authors is to show convergence of adaptive methods for nonconvex optimization. And what I was questioning is that, for the goal of showing convergence in the nonconvex setting, the specialized techniques for improving regret for strongly convex optimization, might not be very insightful.

---

> > > > > ### Author Response · Authors · 2019-11-15
> > > > > **Response to Review #1**
> > > > >
> > > > > We thank the reviewer for the response.
> > > > >
> > > > > Q1: However, given that ICLR is a specialized conference (compared to ICML or Neurips), assuming a very strong structure, which does not hold in many applications that ICLR community is interested in, causes me to justify the suitability of the paper for the conference.
> > > > >
> > > > > A1: We understand that ICLR is more related to representation learning/deep learning, and thus performed experiments on deep neural networks (including a 4-layer CNN and ResNet-18) to examine the empirical performance of SAdam. We note that Mukkamala & Hein (2017) also applied SC-RMSprop to neural networks, and obtain promising results. So, the idea of using a faster decaying step size, although originally designed for strongly convex functions, could lead to superior practical performance even in some highly non-convex cases such as deep learning tasks.
> > > > >
> > > > >
> > > > > A2: We appreciate the insightful suggestion, and will investigate the performance of SAdam in non-convex optimization in future work.

---

### Decision · Program_Chairs · 2019-12-19

**Decision:**

Accept (Poster)

**Comment:**

The reviewers all appreciated the results. They expressed doubts regarding the discrepancy between the assumptions made and the reality of the loss of deep networks.

I share these concerns with the reviewers but also believe that, due to the popularity of Adam, a careful analysis of a variant is worthy of publication.